# Reversible displacive transformation in MnTe polymorphic semiconductor

Shunsuke Mori [1], Shogo Hatayama [1], Yi Shuang [1], Daisuke Ando [1] & Yuji Sutou [1]*

Displacive transformation is a diffusionless transition through shearing and shuffling of atoms. Diffusionless displacive transition with modifications in physical properties can help manufacture fast semiconducting devices for applications such as data storage and switching. MnTe is known as a polymorphic compound. Here we show that a MnTe semiconductor film exhibits a reversible displacive transformation based on an atomic-plane shuffling mechanism, which results in large electrical and optical contrasts. We found that MnTe polycrystalline films show reversible resistive switching via fast Joule heating and enable nonvolatile memory with lower energy and faster operation compared with conventional phase-change materials showing diffusional amorphous-to-crystalline transition. We also found that the optical reflectance of MnTe films can be reversibly changed by laser heating. The present findings offer new insights into developing low power consumption and fast-operation electronic and photonic phase-change devices.

[1] Department of Materials Science, Graduate School of Engineering, Tohoku University, 6-6-11 Aoba-yama, Aoba-ku, Sendai 980-8579, Japan.
*email: ysutou@material.tohoku.ac.jp

Displacive transformation is a diffusionless transition through shearing and shuffling of atoms. Such transformation, commonly known as a martensitic-type transformation, is frequently observed in steels, nonferrous alloys, and ceramics[1,2]. This transformation not only leads to hardening and toughening of materials but also adds functionalities such as changes in physical properties (e.g., resistivity, magnetic susceptibility, and elastic modulus), shape memory properties, and the caloric effect. As this type of transformation does not require random diffusion of atoms, its speed is generally very high; e.g., the speed of martensitic-type transformations approaches the speed of sound (~1000 m s$^{-1}$) in solids. High-speed displacive transition with modifications in physical properties can help manufacture fast semiconducting devices for applications such as data storage and switching. However, no semiconducting devices that operate using displacive transformation have yet been realized.

Polymorphism is of great interest in many fields, including medicine, materials science, catalysis, and electronics, because different forms exhibit dramatically different physical and chemical properties[3]. MnTe is a well-known polymorphic compound[4,5]. Under normal atmospheric conditions, α-MnTe with an NiAs-type hexagonal (NC-type) structure is the most stable phase. In the phase diagram of MnTe, α-MnTe shows structural changes with the increase in temperature as follows: from the β-phase (wurtzite-type hexagonal [WZ-type]) to the γ-phase (sphalerite [zinc-blende]), followed by the δ-phase (rock salt). α-MnTe is a p-type antiferromagnetic semiconductor with a small bandgap of 1.37~1.51 eV[6], whereas β-MnTe is a p-type semiconductor with a wide bandgap of about 2.7 eV[7]. This suggests that MnTe can exhibit a drastic change in electrical and optical properties through polymorphic change.

In this study, we demonstrated that a MnTe semiconductor film that exhibits reversible displacive transformation enables polymorphic change, which results in large electrical and optical contrasts. We found that MnTe polycrystalline films show reversible resistive switching via fast Joule heating and enable nonvolatile memory with a lower energy and faster operation compared with conventional Ge-Sb-Te phase-change materials (PCMs) showing diffusional amorphous-to-crystalline phase transition. Our transmission electron microscopy (TEM) results revealed that an electrical contrast is induced by polymorphic change through displacive transformation based on an atomic-plane shuffling mechanism. The low-resistance α-phase (NC-type structure) transforms into a high-resistance phase with a WZ-type structure (β′-phase), which exhibits slight differences in the coordinates of the Te atoms in the β-phase. Shuffling occurs every two planes along the [210] and [$\overline{2}$10] directions in the α-phase or the β′-phase, which corresponds to the minimum displacements to achieve the polymorphic change between the sixfold-coordinated NC-type and fourfold-coordinated WZ-type structures. Furthermore, we found that the optical reflectance of MnTe films can be reversibly changed by laser heating, indicating that displacive transformation also leads to an optical contrast. The present MnTe polymorphic-change film can be used in a single-polycrystalline structure fabricated by simple, conventional sputtering techniques. This material, therefore, should be a promising material as a memory layer in not only nonvolatile memory but also photonic memory[8,9], nanodisplays[10,11], and neuromorphic devices[12,13].

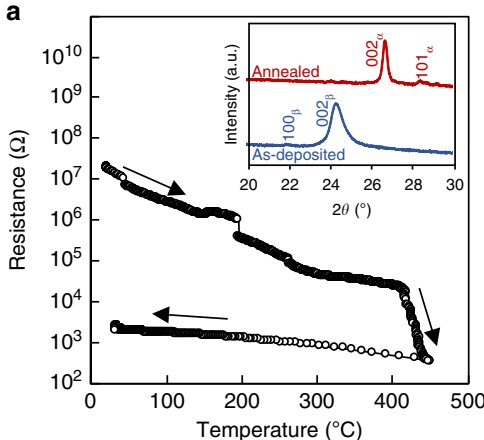

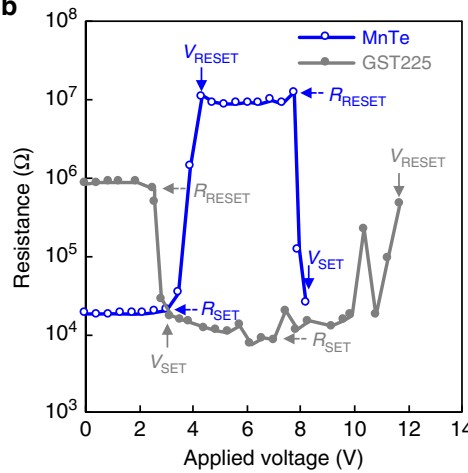

**Fig. 1 Resistance change behavior upon phase change in MnTe.**
**a** Temperature dependence of resistance of the as-deposited MnTe film during heating followed by cooling to RT; the step decrease in resistance at around 200 °C is due to the experimental error caused when the measuring range is switched from 10$^6$ to 10$^5$. The inset shows X-ray diffraction (XRD) patterns at room temperature (RT) of as-deposited (blue) and 500 °C annealed (red) films. **b** A resistance (R) vs. voltage (V) curve for MnTe (open circles) and Ge$_2$Sb$_2$Te$_5$ (GST225) devices (filled circles), read voltage was 0.1 V. The as-annealed MnTe device with low resistance was used for the measurement; the GST225 device was first annealed at 260 °C then cooled to RT and finally it was RESET-operated by applying 13.8 V for 50 ns before the measurement. Solid arrows indicate the critical voltage for RESET ($V_{RESET}$) and SET ($V_{SET}$) operations, and dashed arrows indicate the resistance of RESET ($R_{RESET}$) and SET ($R_{SET}$) states; these parameters were used for the operation energy calculation.

## Results

**Resistance change behavior of MnTe films.** Using electrical resistance measurements as a function of temperature and X-ray diffraction (XRD), we found that a sputter-deposited MnTe film exhibits a large electrical resistance drop at around 435 °C, which is caused by a structural transition from the β-phase (WZ-type structure) to the α-phase (NC-type structure) (Fig. 1a). We also confirmed via XRD measurements in the 2θ range from 20° to 55° that the as-deposited MnTe without a W cap layer has a β-phase (see Supplementary Fig. 1 and Note 1). This polymorphic transition also induced a drastic increase in light absorption[14]. Our TEM results confirmed that the as-deposited film has a β-phase and that, after annealing at 500 °C, an α-phase was obtained (see Supplementary Fig. 2 and Note 2). The lattice parameters of each phase were similar to the reported values[7,15]. Based on the Hall-effect measurements at room temperature (RT), we also found that the drastic resistivity drop upon polymorphic change is mainly caused by a drastic increase in the carrier density (see Supplementary Table 1). Our optical measurements indicated

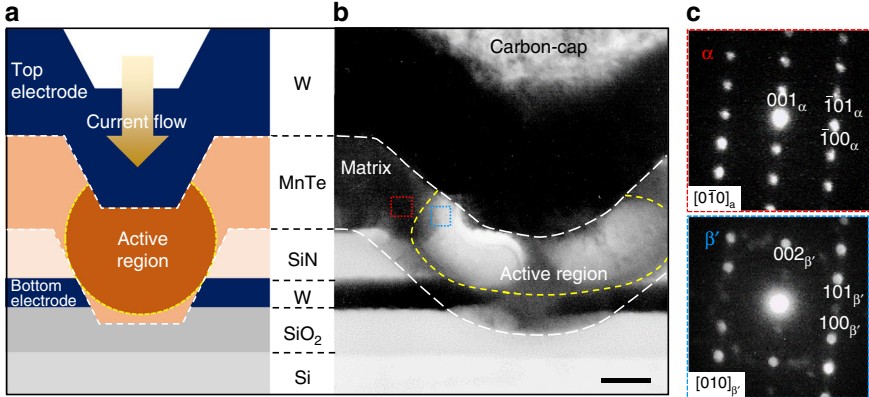

**Fig. 2 Microstructure of the MnTe device after RESET operation. a** Schematic of the cross-section of the MnTe device. **b** Cross-sectional transmission electron microscopy (TEM) image of a MnTe device. Scale bar: 100 nm. The MnTe device showed a resistance switching from low ($5.6 \times 10^3$ Ω) to high ($1.4 \times 10^5$ Ω) resistance state by RESET operation (2.9 V for 50 ns). The bright contrast area in the MnTe layer corresponds to the severely thermally affected active region that was preferentially milled during TEM sample preparation with focus ion beam. **c** Selected area diffraction patterns (SADPs) from the matrix enclosed by the red-dotted line (upper) and active region enclosed by the blue-dotted line (bottom).

that the NC-type α-phase (1.48 eV) has a narrower bandgap compared with the WZ-type β-phase (2.50 eV) (see Supplementary Table 1). Siol et al.[16] have recently calculated the density of states of Mn chalcogenides, such as MnTe, MnSe, and MnSe$_{0.5}$Te$_{0.5}$, with different structures. They demonstrated that the bandgap of a fourfold-coordinated WZ-type structure (2.71 eV) is wider than that of a sixfold-coordinated NC-type structure (0.98 eV) because of the weaker hybridization between the Mn $d$-states and Te $p$-states in the fourfold coordination[16]. That is, the drastic decrease in resistance upon the phase transition from the β-phase to the α-phase is considered to be due to an increase in the coordination number of Mn (or Te) atoms from 4 to 6, which strengthens the hybridization between the Mn $d$-states and Te $p$-states.

**Resistance change behavior of MnTe devices**. The above findings motivated us to investigate the resistive switching behavior of MnTe films via fast Joule heating (see Supplementary Note 3) A resistance ($R$) vs. voltage ($V$) curve under a 50 ns-wide voltage pulse for a MnTe device is shown in Fig. 1b, in which a W layer was used for the top and bottom electrodes (see Supplementary Fig. 3). For comparison, the $R$–$V$ curve for a typical phase-change memory device using Ge$_2$Sb$_2$Te$_5$ (GST225) PCM is also shown in Fig. 1b. Here we observed a resistive switching behavior in the MnTe device, as with the GST225 device. The initial low-resistance state in the MnTe device switches to a high-resistance state when a voltage of $V_{RESET} = 4.3$ V is applied (the RESET operation). Subsequently, a decrease in resistance back to the initial low value is induced by applying a voltage of $V_{SET} = 8.2$ V (the SET operation). We confirmed that the MnTe device shows resistive switching even under a 10 ns-wide voltage pulse, indicating that the MnTe film has a fast operation speed (see Supplementary Fig. 4). This fast operation speed of the MnTe device is almost comparable to that of HfO$_2$ resistive switching memory (~5 ns)[17]. Contrastingly, the GST225 device did not show a SET operation under the 10 ns voltage pulse width (see Supplementary Fig. 5).

It is worth noting that, contrary to the GST225 device, $V_{RESET}$ is lower than $V_{SET}$ in the MnTe device, indicating that the RESET operation energy is very small. Using the $R$–$V$ results, we calculated the total operation energy, $Q_{total}$ (RESET operation energy ($Q_{RESET}$) + SET operation energy ($Q_{SET}$)) for both devices on the basis of Joule's law (see Supplementary Note 4). The $Q_{RESET}$ value of the MnTe device (46.2 pJ) under 50 ns voltage

pulse operation was approximately one order of magnitude lower than that of the GST225 device (794.4 pJ), resulting in less than one-tenth of $Q_{total}$ in the MnTe device compared with the GST225 one (see Supplementary Table. 2). It has been reported that the operation energy strongly depends on the size of the contact area between the memory layer and the electrode, and this energy drastically decreases with the decrease of the size of that area[18]. This indicates that the operation energy of the MnTe device can be reduced by scaling the size of the contact area. Such fast, reversible, low-energy operation implies that the MnTe device exhibits a melting-free, fast, reversible polymorphic change between two crystalline states. In addition, we confirmed that the resistive switching of the MnTe device can be repeated over 400 times (see Supplementary Fig. 6). This cyclic durability is very limited compared with that observed in other memory devices (often > 100,000)[19]. We found that the Mn in the MnTe layer diffuses onto the surface through cracks at the contact hole of the top W electrode and reacts with O to form MnO$_x$ on the surface during cyclic operations. The diffusion of Mn out to the surface leads to the formation of MnTe$_2$ and voids in the MnTe layer, which degrades the cyclic durability (see Supplementary Fig. 7). The aforementioned cracks were unintentionally introduced during the deposition of the W layer, because the contact hole has a concave shape. Further studies are needed to evaluate the cyclic durability of the MnTe film using a device structure without any cracks.

**Phase-transition mechanism**. To understand the mechanism of resistive switching, we directly observed the microstructure of the active region in the MnTe device after the RESET operation. In this experiment, the as-fabricated MnTe device was annealed at 500 °C for 10 min to obtain a low-resistance α-phase with a large grain size and then it was switched to a high-resistance RESET state, enabling the observation of the boundary between the matrix and the active region within a grain of the α-phase (see Supplementary Note 5). A cross-sectional TEM image is shown in Fig. 2. We used the selected area diffraction pattern (SADP) technique to confirm that both the matrix (Fig. 2c, top) and the active region (Fig. 2c, bottom) have crystalline phases. The lattice parameters of the α-matrix with an NC-type structure were $a = 4.225$ Å and $c = 6.785$ Å. When the active region was assumed to have a WZ-type structure, its lattice parameters were $a = 4.211$ Å and $c = 6.806$ Å, smaller than those of the sputter-deposited β-phase (see Supplementary Note 2). This phase, induced by fast

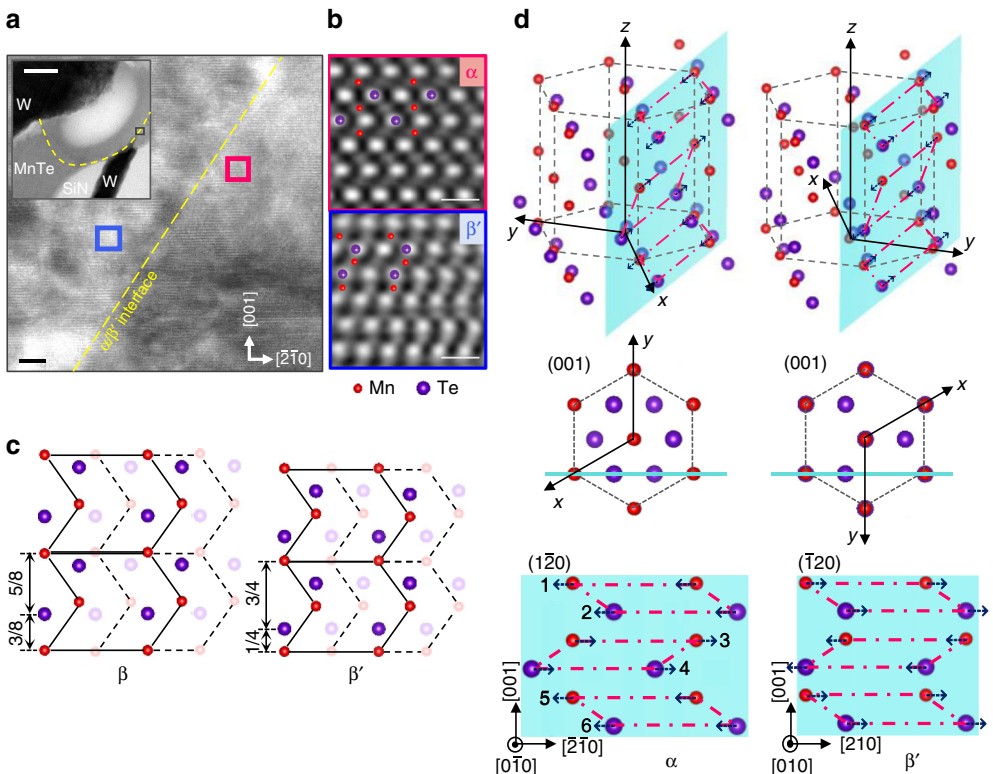

**Fig. 3 Polymorphic-change mechanism in the MnTe device. a** High-resolution transmission electron microscopy (HR-TEM) image in the [0$\bar{1}$0] zone axis of the α-matrix from the area enclosed by the black line in the upper left inset. Scale bar: 5 nm. The upper left inset corresponds to the scanning transmission electron microscopy (STEM) image of Fig. 2b, where the TEM sample was additionally ion milled. Scale bar: 100 nm. **b** Inverse fast Fourier transform (IFFT) images of annular dark-field STEM (ADF-STEM) images of the α (upper)- and β′ (bottom)-phases from the areas enclosed by red (right side) and blue (left side) lines, respectively. Scale bar: 5 Å. **c** Projection of atoms in the β (left)- and β′ (right)-phases viewed from the [010] direction, solid atoms are in the ($\bar{1}$20) plane, and semi-transparent atoms in the neighboring plane. **d** Polymorphic-change mechanism between the α- and β′-phases through alternate atomic-plane shuffling. Upper images show a crystal structure of the α (left)- and β′ (right)-phases. Middle images show a projection of atoms in the α (left)- and β′ (right)-phases viewed from the [001] direction. Bottom images present a projection of atoms on the light-blue colored plane in the α (($\bar{1}$20), left)- and β′ ((($\bar{1}$20)), right)-phase structure. The dotted arrows indicate the shuffling direction of each atomic-plane for the polymorphic-change from the α- to β′-phase (left) and from the β′- to α-phase (right).

Joule heating, is called the β′-phase in this work. In addition, the SADPs also indicated that the *c*-axes in both phases were parallel to each other. To determine the crystal structure of the β′-phase and to understand the polymorphic-change mechanism, we obtained atomic-column images from the area near the α-matrix and the β′-phase boundary using high-resolution TEM (HR-TEM) equipped with annular dark-field scanning TEM (ADF-STEM). An HR-TEM image of the area near the boundary in the [0$\bar{1}$0] zone axis of the α-matrix is shown in Fig. 3a. The corresponding inverse fast Fourier transform (IFFT) images of the ADF-STEM images from the areas enclosed by the red (the α-matrix) and blue (the β′-phase) lines are also shown at the top and bottom of Fig. 3b, respectively. Here, a vague diagonal boundary from the bottom left to the top right was found during the IFFT analysis of the HR-TEM images in various areas (see Supplementary Fig. 8). In these ADF-STEM images, the dark and bright spots correspond to the Mn and the Te columns, respectively; simulated ADF-STEM images viewed from the [010] direction using the multislice method in QSTEM software[20] are shown in Supplementary Fig. 9. First, we found slight differences in the coordinates of the Te atoms between the β-phase and the β′-phase, i.e., the coordinates for the Te atoms along the *z*-axis were (1/3, 2/3, and 3/8) in the β-phase and (1/3, 2/3, and 1/4) in the β′-phase (Fig. 3c). Furthermore, the Mn atoms formed a rectangular lattice on the (1$\bar{2}$0) plane of the α-phase, whereas in the β′-phase the Mn-atom layers were alternately shuffled in the

direction perpendicular to the *c*-axis. This indicated that there is displacive transformation between the α-phase and the β′-phase, as schematically shown in Fig. 3d. The polymorphic change between the α-phase and the β′-phase can be explained by the atomic-plane shuffling parallel to the basal plane with slight expansion/shrinkage along the *a*- and the *c*-axis (below 1%), based on the following orientation relationships:

$$(001)_\alpha \parallel (001)_{\beta'} \text{ and } [\bar{2}\bar{1}0]_\alpha \parallel [210]_{\beta'}, \quad (1)$$

Atomic-plane shuffling occurs every two planes along the [210] and [2$\bar{1}$0] directions in the α-phase or the β′-phase (i.e., atomic planes 1 and 2, and atomic planes 5 and 6 shift by $\sqrt{3}a/6$ in the [210] direction, whereas atomic planes 3 and 4 shift by $\sqrt{3}a/6$ in the [2$\bar{1}$0] direction during the polymorphic change from the α-phase to the β′-phase (Fig. 3d, bottom left). The process of change from the β′-phase to the α-phase occurs in a reversed manner (Fig. 3d, bottom right). This transition pathway corresponds to the minimum displacements, to achieve the polymorphic change between the sixfold-coordinated NC-type and the fourfold-coordinated WZ-type structures, which causes a large resistance contrast. Hence, it is worth investigating the origin of displacive transformation in the MnTe layer. Kurdyumov et al.[21] reported that a Boron Nitride ceramic shows a displacive transition via atomic-plane shuffling along the [001] or the [00$\bar{1}$] direction from

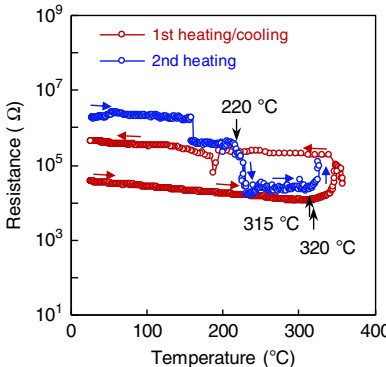

**Fig. 4 Temperature dependence of resistance in the MnTe device.** The initial state of the device was a low resistance state with an α-phase. The MnTe device was annealed up to about 359 °C and then cooled to room temperature (RT) (first heating/cooling cycle). The MnTe device was reheated up to 335 °C (second heating). The step decrease in resistance at around 160 °C in the second heating is due to the experimental error caused when the measuring range is switched from $10^6$ to $10^5$.

a graphite-type hexagonal to a WZ-type hexagonal under high pressure[21]. According to another study[22], the thermal expansion of α-MnTe is remarkable, with 27.9 p.p.m. °C$^{-1}$ and 26.8 p.p.m. °C$^{-1}$ along the *a*- and the *c*-axis, respectively, which are much larger than those of the W (~4.5 p.p.m. °C$^{-1}$) and SiN (1~4 p.p.m. °C$^{-1}$)[23] layers surrounding the MnTe layer in our device. This implies that compressive stress is generated in the α-MnTe layer via fast Joule heating because of the large thermal expansion difference between them. We confirmed that the MnTe device shows a phase transition from the α-phase to the β′-phase even when heated in a furnace chamber (Fig. 4). The MnTe device with an α-phase shows a resistance increase at around 320 °C. Such a resistance increase at around 300 °C was confirmed in three different MnTe devices (see Supplementary Fig. 10 and Note 6). The high resistance was maintained even when the devices were cooled down to RT. We observed the cross-sectional TEM microstructure of the device annealed up to 370 °C followed by air-cooling to RT (see Supplementary Fig. 10). The β′-phase was confirmed to exist near the contact area in the MnTe device. These results strongly support the notion that thermal-stress-induced displacive transformation occurs from the α-phase to the β′-phase, and that the induced β′-phase can be quenched at RT. The expected strain generated in the MnTe layer in the contact hole due to the thermal expansion difference between the MnTe and the surrounding materials is estimated to be about 0.7% (see Supplementary Note 7). The high-resistance state after the first heating/cooling cycle goes back to the initial low-resistance state at around 220 °C, indicating that the induced β′-phase can be transformed back to the thermodynamically stable α-phase through displacive reverse transformation. We also confirmed that the MnTe device that was RESET-operated by an electrical pulse shows a resistance drop at 227 °C during postheating (see Supplementary Fig. 11 and Note 8). This transition value is very similar to that observed in the MnTe device heated in a furnace chamber. Moreover, by further heating the MnTe device, the resistance increases at around 315 °C again, indicating that the transition between the α-phase and the β′-phase is reversible. We also investigated the thickness dependence of the transformation temperature from the α-phase to the β′-phase using 50, 100, 200, and 500 nm-thick α-MnTe films with patterned W electrodes on the surface (see Supplementary Fig. 12 and Note 9). The 100 nm-thick MnTe film showed a slight increase in resistance at around 300 °C, which is very similar to the MnTe device (Fig. 4). We

found that there is negligible correlation between thickness and the transformation temperature in the MnTe films with a thickness more than 100 nm, whereas the transformation temperature tended to decrease slightly with decreasing film thickness in the range of <100 nm. However, the temperature of the 50 nm-thick MnTe film was still high (280 °C). These results suggest that the memory performance depends on the device structure (e.g., memory layer thickness and contact size), as the MnTe film shows thermal-stress-induced phase transformation.

As mentioned above, the transition temperature from the metastable β′-phase to the stable α-phase in the MnTe device was higher than that of the amorphous-to-crystalline phase change in conventional PCMs, 227 °C vs. ~160 °C in GST225, indicating that the MnTe device has an adequate data retention ability (see Supplementary Fig. 11). We confirmed that the metastable β′-phase obtained via the RESET operation can be maintained at 200 °C for over 1 h. This thermal stability was much better than that estimated (for ~15 min at 200 °C) based on the Ozawa method[24] using the activation energy (2.02 eV) of the transition from the β-phase to the α-phase. From the estimation, the MnTe memory layer was estimated to show a 10-year lifetime at a maximum temperature of 103 °C. These results indicate that the MnTe device shows better data retention compared with the conventional GST225 device[25] (over 10 years at 85 °C) (see Supplementary Fig. 13 and Note 10).

**Phase transformation by laser irradiation**. We also demonstrated that the MnTe film capped with a SiO$_2$ layer has a ~20% optical contrast when irradiated with a laser for 80 ns (Fig. 5a). In our experiment, the reflectance of the α-MnTe film abruptly decreased by about 25% with irradiation of 55 mW. The obtained low reflectance started to increase when the film was irradiated with 22 mW and the original high reflectance was nearly recovered using 44 mW irradiation. The reflectance decreased again with laser irradiation of 55 mW. This indicated that there was reversible displacive transformation between the high-reflectance α-phase and low-reflectance β′-phase induced by laser heating. A cross-sectional TEM image of a laser-irradiated MnTe film is shown in Fig. 5b. We confirmed that the phase transition from the α-phase to the β′-phase is induced by laser irradiation (Fig. 5c). These results also support the notion that displacive transformation is thermal-stress-induced, not electric-field-induced. We confirmed that the reflectance of the β-MnTe film on the SiO$_2$/Si substrate is lower than that of the α-MnTe film on the SiO$_2$/Si substrate in the wavelength range between 800 and 900 nm, whereas in the other wavelength range, the α-MnTe film shows lower reflectance compared to the β-MnTe film. The transmittance was higher in the β-MnTe film on a glass substrate than in the α-MnTe film on a glass substrate for the entire wavelength range (see Supplementary Fig. 14 and Note 11).

## Discussion
As promising materials enabling low-energy and fast-operation memory, melting-free PCMs such as GeTe/Sb$_2$Te$_3$ superlattice[26,27] and two-dimensional (2D) In$_2$Se$_3$ polymorph[28,29] have been reported. Moreover, phase-change memory using structural phase transition driven by electrostatic doping has been demonstrated in monolayer MoTe$_2$[30]. However, the fabrication of superlattice and single-crystal 2D or monolayer materials requires complicated processing. Zhang et al.[31] reported that the structural transition from a metastable cubic to a stable hexagonal phase in a crystalline GST225 is induced by shearing martensitic transformation, but the direct reversible phase change via a pure displacive transition has not been demonstrated yet[31,32]. In contrast, the present MnTe

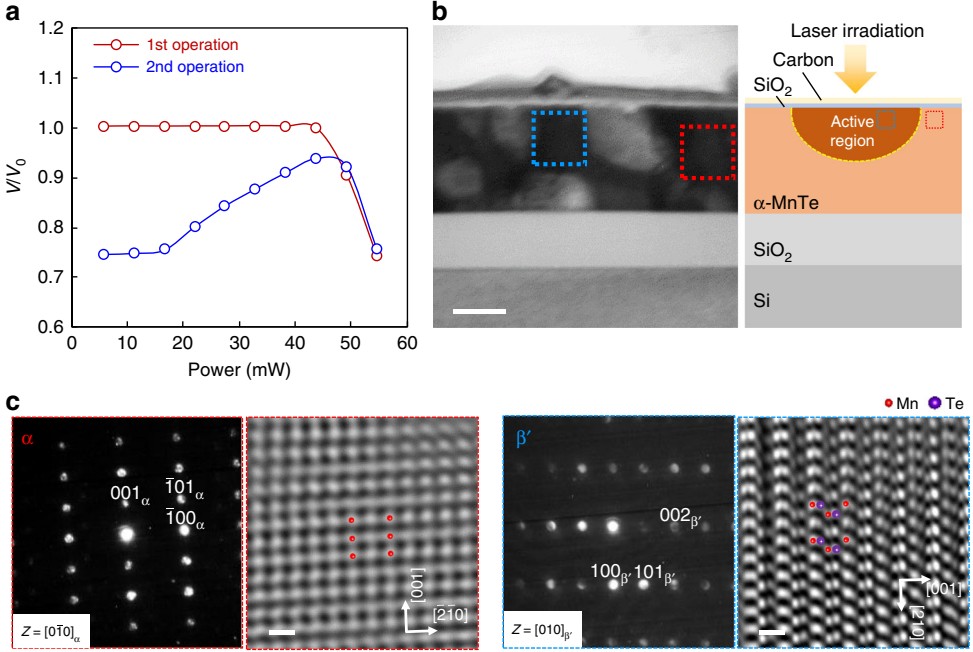

**Fig. 5 Reflectance change in the MnTe film induced by laser irradiation. a** Reflectance change behavior in the MnTe film, induced by laser irradiation, as a function of pump laser power. $V/V_0$ corresponds to the normalized reflectance detected by a photodiode. First, the α-MnTe film was irradiated with the pump pulse laser at the same position and its power was gradually increased. After the reflectance dropped, the laser power returned to zero and then again gradually increased, irradiating at the same position than the first measurement. **b** Cross-sectional transmission electron microscopy (TEM) image of the MnTe film that showed a reflectance decrease by laser irradiation. Scale bar: 100 nm. Schematic of the cross-section of the MnTe film is shown in right inset. Only a small part near surface is supposed to be active region. **c** Selected area diffraction patterns (SADPs) and inverse fast Fourier transform (IFFT) images of high-resolution TEM (HR-TEM) images taken from α-matrix enclosed by the red-dotted line (left) and active region enclosed by the blue-dotted line (right). Scale bar: 5 Å. The active region exhibits a β'-phase. In the IFFT images, both Mn and Te atoms were observed in the β'-phase, whereas only Mn atoms were visible in the α-phase. Both images were well fitted with the simulation results at a film thickness of $t = 9.9$ nm and defocus $\Delta f = +2$ (see Supplementary Fig. 8a, b).

polymorphic-change film can be used in single-polycrystalline structures fabricated by conventional and simple sputtering techniques, and, therefore, should be a promising material as a memory layer in nonvolatile random access memory. To realize such a hypothesis, we need further studies on the optimal memory cell structure, optimal electrode material, and optimal selector material for the MnTe memory layer. In addition, the MnTe film showing thermal-stress-induced polymorphic change should be a very attractive material for straintronics in which a phase transition induced by mechanical strain causes a change in electrical or optical properties[33,34]. Moreover, optically driven and fast martensitic transition has been recently theoretically and computationally illustrated in 2D materials, such as SnO and SnSe[35]. The present findings of the nonvolatile, reversible displacive transformation in the semiconductor chalcogenide may also lead to the materialization of optically driven, exceptional, fast nonvolatile memory.

## Methods

**Preparation of MnTe films**. MnTe films (200 nm thick) were deposited on Si (725 μm)/SiO$_2$ (100 nm) or glass (Corning EAGLE XG) substrates by radiofrequency (RF) magnetron cosputtering of Mn and Te pure targets at RT, where the substrate holder was rotated during deposition. The diameter and thickness of the targets were 50.8 mm and 5 mm, respectively. The distance between the target and the substrate was about 300 mm. The RF power used for sputtering deposition was 40–50 W for the Mn target and 10 W for the Te target. The base pressure of the sputtering chamber was below $5.0 \times 10^{-5}$ Pa and the Ar gas was introduced at 15 s.c.c.m. for film deposition. The working pressure was around $3.6 \times 10^{-1}$ Pa. During the film deposition, the substrate was not intentionally heated and no bias was applied to the substrate. The substrate temperature was monitored using a thermocouple attached to the substrate holder and was confirmed to be below 300 K (27 °C) during deposition. Subsequently, to prevent surface oxidation of the MnTe films during annealing, a 100 nm-thick W cap layer was deposited on the

films by sputtering in the same chamber without breaking the vacuum. For static laser testing, a 5 nm-thick cap layer of SiO$_2$ was used instead of the W cap layer. This SiO$_2$ cap layer was deposited by sputtering of a SiO$_2$ target and the film was then annealed at 500 °C and cooled to RT to obtain an α-phase. To control the film thickness, we first evaluated the sputtering rates of the targets by measuring the thicknesses of the films deposited for a given time using atomic force microscopy (AFM) (VN-8000; KEYENCE). Before deposition, the substrate surface was partially covered with a marker and the film was then deposited on the substrate. After removing the marker, we measured the step (i.e., the film thickness) with AFM. A 1 μm-thick MnTe film without a W cap layer was also prepared to determine the crystal structure of the as-deposited film.

**MnTe film characterization**. The temperature dependence of the resistance of the MnTe film was investigated using the two-point probe method in the range of 20 °C–500 °C under an Ar atmosphere with a heating rate of about 10 °C min$^{-1}$. In this measurement, a 200 nm-thick W layer was also deposited to prevent the surface oxidation of the MnTe film, but this W layer was divided into two segments using a photolithographic technique to measure the resistance change behavior of the MnTe film. The distance between the two segments was 80 μm and the remaining MnTe surface was covered with a nonconductive carbon layer. It is worth noting that the C cap layer was also deposited on the W electrode, but it was easily removed by simply placing the probe electrode and therefore the probe was in direct contact with the W electrode.

The dependence of the heating rate on the transition temperature of the as-deposited MnTe film was investigated using the two-point probe method at various heating rates of 4.6–23 °C min$^{-1}$ and the activation energy of the transition was estimated using the Kissinger method.

The transmittance and reflectance of the sputter-deposited MnTe film before and after annealing were measured using a spectrophotometer (V-630 BIO; JASCO) in the 400–1100 nm range. The reflectance of the film was measured with respect to the reflectance of an Al reference mirror. The sputter-deposited MnTe film was annealed at 500 °C followed by cooling to RT without any holding time at 500 °C. In this measurement, a C cap layer was used. From these results, we evaluated the optical bandgap on the basis of the Tauc equation. We also measured the electrical resistivity, Hall mobility, carrier concentration, and carrier type of the MnTe film at RT using a Hall-effect measurement system (Resi Test 8403; TOYO Corporation) with W probe electrodes. In both experiments, the thickness of the

MnTe film was about 100 nm. The results of the optical and the Hall-effect measurements are summarized in Supplementary Table S1. The microstructure of the MnTe film was observed using field-emission TEM (FE-TEM) (HF-2000EDX; HITACHI) at an accelerating voltage of 200 kV.

**Memory device fabrication.** To evaluate the resistive switching behavior of the MnTe film, a MnTe memory device was fabricated using conventional photo-lithography combined with focus ion beam (FIB) drilling. First, a 50 nm-thick W layer was deposited on a Si (725 μm)/SiO$_2$ (100 nm) substrate as the bottom electrode layer using photolithography. A 100 nm-thick SiN insulating layer was then deposited using N$_2$ gas reactive sputtering of a pure Si target on the W bottom layer, also using a photolithography pattern. Subsequently, a contact hole (500 nm diameter) was fabricated with FIB drilling. A 200 nm-thick MnTe layer was then deposited onto the contact hole, followed by the deposition of a W top electrode layer of the same thickness without breaking the vacuum. Finally, the device was annealed at 500 °C, followed by cooling to RT with no holding time at 500 °C. A MnTe layer with an α-phase was obtained. For comparison, a Ge$_2$Sb$_2$Te$_5$ (GST225) device was also fabricated using the same process, with annealing at 260 °C to obtain a crystalline phase. A schematic of the device is shown in Supplementary Fig. 3.

In addition, we prepared 50, 100, 150, and 200 nm-thick α-MnTe films with a patterned W electrode on the surface to investigate the dependence of the film thickness on the transition temperature of the MnTe film. First, we formed patterned W electrodes on as-deposited β-MnTe films. Next, the β-MnTe films were annealed up to 550 °C and showed a transition to a low-resistance α-phase (see Supplementary Fig. 12a).

**Memory device characterization.** The resistive switching properties of the memory devices were evaluated using a semiconductor parameter analyzer (Keysight B1500A). A short voltage pulse (width: 30–50 ns) was provided by a pulse generator (Keysight B1525A), with an output impedance of 50 Ω. The pulse voltage applied to the memory device was verified with an oscilloscope (DSO3062A from Agilent Technologies or TBS 1202B from Tektronix). The resistive switching operation under a voltage pulse with a minimum width of 10 ns (machine limit) was also evaluated using a different pulse generator (33250A; Agilent Technologies), with an output impedance of 50 Ω.

The temperature dependence of the resistance of the MnTe devices and the MnTe films with patterned W electrodes was also investigated using the two-point probe method in the range of 20 °C–370 °C under an Ar/5% H$_2$ mixed gas atmosphere with a heating rate of about 10 °C min$^{-1}$.

The cross-sectional microstructure of the MnTe device after the RESET operation (i.-e., low- to high-resistance state) was observed using FE-TEM (HF-2000EDX; Hitachi) at an accelerating voltage of 200 kV. To obtain atomic images, we used HR-TEM equipped with an ADF-STEM detector (ARM200F; JEOL). IFFT images were obtained using a Gatan DigitalMicrograph®. TEM samples were thinned using ion milling (PIPS; Gatan) or FIB (JIB-4600F; JEOL). To observe the active region in the MnTe device, the as-fabricated MnTe device was annealed at 500 °C for 10 min, followed by cooling to RT to obtain an α-phase at a low-resistance state with a large grain size (~250 nm); it was then reset by applying 2.9 V for 50 ns before the preparation of the TEM sample. By enlarging the grain size of the α-matrix when annealing at 500 °C, we could observe the boundary between the α-matrix and the active region within the grain. During the RESET operation, the MnTe device switched from low (5.6 × 10$^3$ Ω) to high (1.4 × 10$^5$ Ω) resistance. STEM energy-dispersive X-ray spectrometry confirmed that there was no interface diffusion between the MnTe and W layers after annealing at 500 °C for 10 min.

To understand the failure mechanism of the MnTe device, we observed its cross-sectional TEM microstructure, which broke down after SET–RESET operation. The TEM sample was thinned using the FIB technique.

**Static laser testing.** We investigated whether the thermal energy due to laser irradiation induced any polymorphic changes, by evaluating the reflectance variation using static laser testing. A pump pulse laser with a constant wavelength of 830 nm was used to provide the thermal energy for polymorphic change. The pulse width of the pump laser was fixed to 80 ns. In this test, the normalized reflectance V/V$_0$ was evaluated, where V$_0$ and V were the voltages detected by a photodiode before and after pump pulse laser irradiation and were proportional to the absolute reflectance. The wavelength of the probe laser was 830 nm.

The cross-sectional microstructure of the MnTe film after laser irradiation was observed using the HR-TEM technique. The MnTe film with an α-phase was irradiated with a pulse laser of 55 mW for 80 ns, causing a decrease in reflectance, and the TEM sample was thinned using the FIB technique.

## Data availability

The data are available from the corresponding authors upon request. The present authors, S.M. and Y. Sutou are inventors on Japanese patent application number 2018-035863, applied for by Tohoku University.

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

## Acknowledgements

This work was supported by the JSPS KAKENHI Grant Number 18H02053 and 19J21117. We wish to thank Kosei Kobayashi, Takamichi Miyazaki, and Masatoshi Tanno, Tohoku University, Japan, for the help with the TEM measurements, and Junichi Koike, Tohoku University, Japan, for fruitful discussion.

## Author contributions

S.M. and Y. Sutou conceived the concepts, designed the experiments, and co-wrote the manuscript with input from all other authors. Y. Sutou led the project. S.M. carried out film deposition, device fabrication, and XRD. S.M. and Y. Sutou performed electrical measurements and static laser testing along with S.H. and Y. Shuang. S.M. and D.A. carried out TEM observations and analysis. S.M., Y. Sutou, S.H., Y. Shuang, and D.A. discussed all the data and their interpretations.

## Competing interests

The authors declare no competing interests.

## Additional information

**Peer Review Information** *Nature Communications* thanks Raffaella Calarco and other, anonymous, reviewer(s) for their contribution to the peer review of this work. Peer reviewer reports are available.

