## [Peer Review File · Nature Communications]

Reviewers' comments:

Reviewer #1 (Remarks to the Author):

Comments regarding novelty and impact:

In their paper "Reversible displacive transformation in MnTe polymorphic-semiconductor" Mori et al. demonstrate a semiconducting device, which utilizes fast displacive polymorphic transitions in MnTe for resistive memory switching applications. The results are intriguing since the study utilizes polymorphism as a functional property, an idea that has gained traction in the materials science community in recent years (Chem. Soc. Rev., 2019, 48, 2502-2517). The fact that the phase transformation is displacive is of particular interest, as it allows for high operating speeds of future devices.

The most novel aspect of the study, in my opinion, is the demonstrated ability to carefully and more importantly, reversibly control polymorphic transitions in MnTe. This is a topic that might be of interest for a broader community, not limited to research on resistive switching memory applications. This result is exciting from a fundamental research point of view, which is why it is very important to rigorously analyse the origin of the observed phase change – an aspect of the manuscript that the authors should improve:

The authors hypothesize on page 4 that the origins of the phase transformations are: a) stress-induced stabilization in the case of the transformation from the alpha to the beta' phase and b) a displacive transformation at high temperature into the thermodynamic ground state in the case of the transformation from the beta' to the alpha phase.

This hypothesis is based on the different thermal expansion coefficients of the contact materials W and SiO₂ compared to MnTe. The authors claim that these different thermal expansion coefficients result in lateral strain upon joule heating, leading to the observed phase transformations.

The hypothesis is further supported by the results of laser heating induced phase transitions without the presence of an electric field. However I don't believe that the laser heating experiment alone provides sufficient experimental proof for this hypothesis. The reflectivity of the film is changing during the phase formation which also influences the effectiveness of the laser heating. This in turn means that it is not clear how the laser power relates to the internal temperature of the device. In addition the laser used in the study has a wavelength of 830nm corresponding to a photon energy of 1.49 eV. While this is barely enough to be absorbed in alpha-MnTe it certainly is below the absorption onset of beta-MnTe (2.5-2.7eV). The laser wavelength is therefore not suitable to effectively heat up the beta-MnTe. It is not surprising that a change in reflectivity is observed given the fact that the beta' phase shouldn't absorb the laser radiation.

An alternative and more accurate way to test the hypothesis of stress induced phase transformations could be to take the device and heat it to different temperatures, i.e. a similar experiment as for the bulk material but for the finished devices. For intermediate temperatures the beta' phase should become stable, whereas for higher temperatures the alpha phase should be come stable. This way it should also be possible to pinpoint the exact transition temperatures. This is particularly interesting given that at high temperatures the thermodynamic driving force towards the ground state structure is large but the strain should also be higher. It can therefore not be expected that the beta-alpha transition happens at the same temperature as observed for the bulk samples.

It might also be interesting to calculate the actual strain, that would be expected upon heating and relate this to the observed phase transition considering the geometry of the device and the mismatch in thermal expansion coefficients. In addition, the authors should discuss the influence of the MnTe layer thickness. If the stabilization is truly stress-induced I would assume that it does

only work up to a critical thickness of the MnTe layer.

General notes regarding the presentation of the results and the motivation of the study:

The authors should give a more thorough overview of the various polymorphs that have been observed in MnTe and discuss the potential formation of a zinc blende phase. This phase has been synthesized using MBE (Chem. Phys. Lett., 2004, 387, 110-115). Looking at the XRD pattern in Figure 1 this structure could also provide a reasonable fit. It is suggested that the range of this XRD pattern is extended to higher angles.

In addition, the authors should provide a more complete description of the state of the art in resistive memory switching research. It is certainly valuable that a comparison with a benchmark material Ge-Sb-Te is provided, but in terms of switching speed, there are certainly better/faster alternatives, such as HfO₂.

With respect to the durability of the devices, concerns can arise if one of the states is constituted of a metastable polymorph. In the present study durability is demonstrated over 400 cycles. Comparable studies using other resistive switching materials demonstrate much higher cycle numbers (often >100000). It would be of interest to investigate why the MnTe-based devices failed after 400 cycles and discuss how this issue could be solved. It is also important, to assess the stability of the beta' phase without cycling. How long can the system remain in the beta' phase considering environmental factors (e.g. temperatures in the range of 70°C-80°C) without showing a transformation into the alpha ground state?

Finally, in the conclusion multiple other developments in the field are listed. Of course this highlights the relevance and impact of the current study, but I would rather expect to find this information in the introduction. Instead I would appreciate if the authors could share their vision on how research on MnTe ReRAM devices should carry on and what needs to be done to make MnTe ReRAM devices a reality.

Comments regarding the experimental reproducibility of the results:

There have been multiple studies on the magnetron sputtering of MnTe. In almost all of those reports (e.g. Reference 6, J. Mater. Chem. C, 2018,6, 6297-6304) the alpha phase (NiAs-Structure) was observed. More details regarding the geometry of the deposition chamber as well as base pressure, power densities should be given. The authors should also provide an estimate/measurement of the effective temperature during deposition since room temperature depositions are always subject to unintentional heating and the polymorph formation can be highly influenced by the substrate temperature during sputtering.

The starting point for the experiments as reported is a beta-MnTe, which is then heated to achieve the ground state alpha-MnTe. This alpha-phase is the thermodynamic ground state and would be the starting point for many other researchers using other sputter systems or near-equilibrium deposition techniques. The authors should discuss the impact of the structural properties of the starting materials. If the MnTe was sputtered at high-temperature (e.g. 500°C) the as deposited material would likely be already in the alpha phase. Does such a material also exhibit the discussed displacive phase transformation behaviour or is the starting point (a metastable state which is then annealed) important for the functionality of the device.

On page 3 a calculation of the total operation energy is carried out. To relate these values to the transition temperatures discussed elsewhere in the manuscript the dimensions of the active region of device should be known. It is my understanding that this calculation gives the total energy whereas the local heating, which is relevant for the phase transformations very much depends on the geometry of the device.

Minor corrections:

Typo on the bottom of page 3. "Frist".

Reviewer #2 (Remarks to the Author):

The work describes the displacive transformation of MnTe in a reversible fashion for the realization of novel generation of non-volatile random-access memories. The authors compare the performances of MnTe with the GeSbTe (GST) alloy used for phase change memories (PCM). The paper contains solid experimental evidence of a reversible displacive transformation. Amazingly enough, MnTe displays low power consumption and fast switching.

The manuscript is well-written. The results are convincing, but for the exceptions noted below, and the insight provided will certainly prove to be of great interest well beyond the phase change material community. Some technical issues are present - also noted below.

Specifically:

- When referring to GST the authors probably refer to the GST225 composition and this should be clarified.
- The cycling of the MnTe device is very low: only 400 cycles. For GST usually cycling is in the order of a million. The authors should explain the possible failure mechanisms and the possible future possibility to improve in that respect. Without the expectation of improving, MnTe is not at all suitable for any kind of memory. Thus, the manuscript does not deserve a publication in a high impact journal.
- The explanation of the displacive transformation is given on page 4, but for helping the readers it would be helpful to summarize the results a bit before. Furthermore, figure 3D is rather difficult to understand, as the shuffling is not very intuitive.
- A recent paper on PCM superlattices (Boniardi et al. Phys. Status Solidi RRL 13 (4), 1800634 (2019) <https://doi.org/10.1002/pssr.201970021>) has been also reporting low power consumption it might be useful to cite such work to compare with it and not with standard GST.

Reviewer #3 (Remarks to the Author):

In the manuscript Authors present results on the reversible resistive switching by fast Joule heating of MnTe polycrystalline films that might be promising for non-volatile memory. Authors compared MnTe with conventional Ge-Sb-Te phase-change materials. The manuscript is well structured and the data is nice and might be interesting for the wide range of readers. However, there are following comments that need to be answered before this manuscript will be in the condition to be accepted for the publication.

1. In GSTs the contrast of the properties (electrical, optical) caused by the difference in the chemical bonding in amorphous and the metastable rocksalt crystalline state. GSTs have stable hexagonal phase whose properties are different from rocksalt crystalline state. Actually, the amplitude of the properties contrast is driven by the interplay of effects of the resonant bonding and the disorder.

In the manuscript Authors demonstrated the difference between different crystalline states of MnTe and therefore claim that this material might be promising for new memory concepts. It is, however, need to be clarified in details what is the origin of the property contrast in different crystalline states of MnTe and to discuss the possible "tools" (like disorder in GSTs) that might manage the amplitude of the properties contrast.

2. To be employed in the memory device on one hand material need to show the stability of the states attributed to the logical "0" and "1", on the other hand need to show reasonable quantity of writing cycles. In case of GSTs, these parameters are in the scales of years and millions of cycles, correspondingly. In the manuscript Authors write about "over 400 switching cycles", that is

definitely not enough for real industrial applications. What is the maximum possible number switching cycles in case of MnTe? What about the stability of corresponding crystalline states attributed to the logical "0" and "1"?

3. If in the Abstract Authors write "...we show a MnTe semiconductor film that exhibits a reversible displacive transformation that results in large electrical and optical contrasts..", I will strongly recommend to improve the part discussing the optical contrast by providing corresponding data in wide spectral range that will definitely support the mentioned above statement.

[Our response to the referees' comments]

Thank you very much for many valuable comments.

In compliance with the reviewer's comments, we conducted additional experiments and considered the following modifications.

The modified parts were highlighted by blue color in the revised main text. Details of our answers for the reviewer's comments are as follows.

Reviewer #1:

Comment 1:

The authors hypothesize on page 4 that the origins of the phase transformations are: a) stress-induced stabilization in the case of the transformation from the alpha to the beta' phase and b) a displacive transformation at high temperature into the thermodynamic ground state in the case of the transformation from the beta' to the alpha phase. This hypothesis is based on the different thermal expansion coefficients of the contact materials W and SiO₂ compared to MnTe. The authors claim that these different thermal expansion coefficients result in lateral strain upon joule heating, leading to the observed phase transformations.

The hypothesis is further supported by the results of laser heating induced phase transitions without the presence of an electric field. However I don't believe that the laser heating experiment alone provides sufficient experimental proof for this hypothesis. The reflectivity of the film is changing during the phase formation which also influences the effectiveness of the laser heating. This in turn means that it is not clear how the laser power relates to the internal temperature of the device. In addition the laser used in the study has a wavelength of 830nm corresponding to a photon energy of 1.49 eV. While this is barely enough to be absorbed in alpha-MnTe it certainly is below the absorption onset of beta-MnTe (2.5-2.7eV). The laser wavelength is therefore not suitable to effectively heat up the beta-MnTe. It is not surprising that a change in reflectivity is observed given the fact that the beta' phase shouldn't absorb the laser radiation.

Authors: Thank you very much for your comment. The reviewer is correct that the β' phase cannot absorb the laser light with a wavelength of 830 nm due to its wider band gap. However, not whole area of film in depth has changed from the α phase to the β' phase, but only a small area near the surface has changed from the α phase to the β' phase by laser irradiation. In this case, the α phase existing around the β' phase can absorb the laser light, causing enough laser heating for a transition back from the β' phase to the α phase. To confirm the phase transition by the laser heating experiment, we directly observed the cross-section of the laser irradiated-part using TEM. The results are indicated in Fig. R1. In this experiment, the α phase film was irradiated with a pulse laser of 55 mW and 80 ns, causing a decrease in reflectance. We confirmed that the laser irradiation induces a transition from the α phase to the β' phase.

Fig. R1. Cross-sectional TEM image of the MnTe film that showed a reflectance decrease by laser irradiation (upper figure). Schematic of the cross-section of the MnTe film is shown in right inset. Only a small part near surface is supposed to be active region. SADPs and IFFT images of HR-TEM images taken from α -matrix enclosed by the red-dotted line (bottom left) and active region enclosed by the blue-dotted line (bottom right). The active region exhibits a β' phase. In the IFFT images, both Mn and Te atoms were observed in the β' phase, while only Mn atoms were visible in the α phase. Both images were well fitted with the simulation results at a film thickness of $t = 9.9$ nm and defocus $\Delta f = +2$ (see Supplementary Fig. 10A and B).

In response to the reviewer's comment, we have added the additional results and revised the text as follows:

1. Additional results of TEM observation were added in old Fig. 4 in the original manuscript (new Fig. 5 in the revised manuscript). And, the caption was revised.
2. We modified the original sentence (p5, line 4th-7th in the original text) as follows:
 "A cross-sectional TEM image of a laser-irradiated MnTe film is shown in Fig. 5B. We confirmed that the phase transition from the α -phase to the β' -phase is induced by laser irradiation (Fig. 5C). These results also support the notion that displacive transformation is thermal-stress-induced, not electric-field-induced." (p8, line 4th-7th in the revised text).
3. The following sentence was added in Methods, Static laser testing:
 "The cross-sectional microstructure of the MnTe film after laser irradiation was observed using the HR-TEM technique. The MnTe film with an α -phase was irradiated with a pulse laser of 55 mW for 80 ns, causing a decrease in reflectance, and the TEM sample was thinned using the FIB technique." (p12, line 2nd-5th in the revised text).

Comment 2:

An alternative and more accurate way to test the hypothesis of stress induced phase transformations could be to take the device and heat it to different temperatures, i.e. a similar experiment as for the bulk material but for the finished devices. For intermediate temperatures the beta' phase should become stable, whereas for higher temperatures the alpha phase should be come stable. This way it should also be possible to pinpoint the exact transition temperatures. This is particularly interesting given that at high temperatures the thermodynamic driving force towards the ground state structure is large but the strain should also be higher. It can therefore not be expected that the beta-alpha transition happens at the same temperature as observed for the bulk samples.

Authors: Thank you very much for your valuable comment. As the reviewer pointed out, we investigated the transition behavior of the MnTe device upon annealing (heating) with heater furnace. We prepared three devices ($N=3$) with a low resistance state of an α phase and then annealed them up to more than 300°C. The results are shown in Fig. R2A. We found that the devices show a resistance increase at around 300°C, indicating that a transition from the α to β' phase occurs. And, even when the devices are cooled down to room temperature (RT), the high resistance state is maintained. We observed the cross-sectional TEM microstructure of the device annealed up to 370°C followed by air-cooling to RT. We confirmed that that the β' phase exists near the contact area in the MnTe device, as shown in Fig. R2B. These results strongly support that a thermal stress-induced displacive transformation occurs from the α to the β' phase and the induced β' phase can be quenched at RT. Here, it is noted that the resistance contrast obtained by heating the MnTe device is smaller than that obtained by local Joule heating, indicating that the local Joule heating generates much larger compressive stress in the contact hole than the furnace heating and therefore, the volume fraction of the β' phase in the contact hole is considered to become large.

Fig. R2. (A) Temperature dependence of the resistance of the MnTe devices ($N = 3$). The initial state of the MnTe layer in the devices was an α -phase. All devices showed an increase in resistance at around 300°C, indicating a transition from the α -phase to the β' -phase. (B) Cross-sectional TEM image of the MnTe device after heating up to 370°C. The inset shows an IFFT image of the HR-TEM image taken from near the contact area enclosed by the blue dotted line.

Moreover, we reheated the MnTe device to evaluate the transition temperature from the β' phase to the α phase. The result is shown in Fig. R3. The high resistance state after the 1st heating/cooling cycle goes back to the initial low resistance state at around 220°C, indicating that the induced β' phase can be transformed back to the thermodynamically stable α phase. This transition temperature from the β' to α phase is in good agreement with the result indicated in supplementary Fig. 8 of the original manuscript which indicates that the β' phase obtained by fast Joule heating transforms to the stable α phase at around 227°C. And, by further heating, the resistance increases at around 320°C again due to a transition from the α to β' phase.

Fig. R3. Temperature dependence of the resistance of the MnTe device. The initial state of the device was a low resistance state with an α phase. The MnTe device shows an increase in resistance at around 320°C in the 1st heating process, while in the 1st cooling process, the high resistance state is maintained. In the 2nd heating process, the resistance drops at around 220°C and increases again at around 315°C.

In response to the reviewer's comment, we have added the additional results and revised the text as follows:

1. Additional results of temperature dependence of resistance in the MnTe device were added in new Fig. 4 in the revised manuscript.
2. We added additional results in Supplementary information, Section 6 and Supplementary Fig. 10.
3. We modified the original sentence (p4, line 24th-28th in the original text) as follows:
 “We confirmed that the MnTe device shows a phase transition from the α -phase to the β' -phase even when heated in a furnace chamber (Fig. 4). The MnTe device with an α -phase shows a resistance increase at around 320 °C. Such a resistance increase at around 300 °C was confirmed in three different MnTe devices (see Supplementary Section 6 and Fig. 10). The high resistance was maintained even when the devices were cooled down to RT. We observed the cross-sectional TEM microstructure of the device annealed up to 370 °C followed by air-cooling to RT (see supplementary Fig. 10). The β' -phase was confirmed to exist near the contact area in the MnTe device. These results strongly support the notion that thermal-stress-induced displacive transformation occurs from the α -phase to the β' -phase and that the induced β' -phase can be

quenched at RT. The expected strain generated in the MnTe layer in the contact hole due to the thermal expansion difference between the MnTe and the surrounding materials is estimated to be about 0.7% (see Supplementary Section 7). The high-resistance state after the first heating/cooling cycle goes back to the initial low-resistance state at around 220 °C, indicating that the induced β' -phase can be transformed back to the thermodynamically stable α -phase through displacive reverse transformation. We also confirmed that the MnTe device that was RESET-operated by an electrical pulse shows a resistance drop at 227 °C during postheating (see Supplementary section 8 and Fig. 11). This transition value is very similar to that observed in the MnTe device heated in a furnace chamber. Moreover, by further heating the MnTe device, the resistance increases at around 315 °C again, indicating that the transition between the α -phase and the β' -phase is reversible.” (p6, line 28th - p7, line 11th in the revised text).

4. The following sentence was added in Methods, Memory device characterization:

“The temperature dependence of the resistance of the MnTe devices and the MnTe films with patterned W electrodes was also investigated using the two-point probe method in the range of 20 °C –370 °C under an Ar/5% H₂ mixed gas atmosphere with a heating rate of about 10 °C/min.” (p11, line 10th-12th in the revised text).

Comment 3:

It might also be interesting to calculate the actual strain, that would be expected upon heating and relate this to the observed phase transition considering the geometry of the device and the mismatch in thermal expansion coefficients.

Authors: Thank you very much for your advice. The transition temperature from the α to β' phase was determined to be around 300°C. Thus, we calculated the expected strain generated by the difference of thermal expansion between the MnTe layer and electrode/insulator layer upon heating at 300°C. We used the thermal expansion value of 27.4 ppm/°C (= (27.9 ppm/°C for a-axis + 26.8 ppm/°C for c-axis)/2) for MnTe and 4 ppm/°C for W electrode/SiN insulator layer. Here, for simplicity, only thermal expansion in the diameter direction of the contact hole is considered. The expected strain is estimated to be about 0.7%.

In response to the reviewer’s comment, we have added the additional results and revised the text as follows:

1. We added the above discussion in Supplementary information, Section 7.

2. We added the above discussion in the revised manuscript as follows:

“The expected strain generated in the MnTe layer in the contact hole due to the thermal expansion difference between the MnTe and the surrounding materials is estimated to be about 0.7% (see Supplementary Section 7).” (p7, line 1st-3rd in the revised text).

Comment 4:

In addition, the authors should discuss the influence of the MnTe layer thickness. If the stabilization is truly stress-induced I would assume that it does only work up to a critical thickness of the MnTe layer.

Authors: Thank you very much for your comment. The reviewer's comment motivated us to investigate the effect of film thickness on the transformation behavior. We prepared 50, 100, 150 and 200-nm thick α -MnTe films with patterned W electrodes on the surface. Then, we investigated the temperature dependence of the resistance of those films. The results are shown in Fig. R4. The 100-nm thick MnTe film shows a slight increase in resistance at around 320°C, which is very similar to the MnTe device, as mentioned in comment 2. The resistance increase in the 100-nm thick film is considered to be caused by a transition from the α to β' phase. The onset temperature at which the resistance starts to increase, indicating the transition temperature from the α to β' phase, is plotted as a function of film thickness in the inset of Fig. R4. The result indicates that the transformation temperature tends to decrease slightly, as the film thickness decreases. The results suggest that the memory performance depend on the device structure, i.e., memory layer thickness, contact size and so on, because the MnTe film shows a thermal stress-induced phase transformation.

Fig. R4. (A) Schematic description of the MnTe film for the investigation of the temperature dependence of the resistance (upper: cross-section, bottom: top view). (B) Temperature dependence of the resistance of the 50, 100, 150, and 200 nm thick α -MnTe films with patterned W electrodes on the surface. The inset shows the thickness dependence of the onset temperature (indicated by arrows), at which the resistance starts to increase, indicating the transition temperature from the α -phase to the β' -phase.

In response to the reviewer's comment, we have added the additional results and revised the text as follows:

1. We added additional results of temperature dependence of resistance in the MnTe film with various film thickness in Supplementary Section 9 and Supplementary Fig. 12 in the revised manuscript.

2. We added the following sentence:

“We also investigated the thickness dependence of the transformation temperature from the α -phase to the β' -phase using 50, 100, 200, and 500 nm thick α -MnTe films with patterned W electrodes on the surface (see Supplementary Section 9 and Supplementary Fig. 12). The 100 nm thick MnTe film showed a slight increase in resistance at around 300 °C, which is very similar to the MnTe device (Fig. 4). We found that there is negligible correlation between thickness and the transformation temperature in the MnTe films with a thickness more than 100 nm, whereas, the transformation temperature tended to decrease slightly with decreasing film thickness in the range of less than 100 nm. However, the temperature of the 50 nm thick MnTe film was still high (280 °C). These results suggest that the memory performance depends on the device structure (e.g., memory layer thickness and contact size) since the MnTe film shows thermal-stress-induced phase transformation.” (p7, line 11th-21st in the revised text).

3. The following sentence was added in Methods, Memory device fabrication:

“Additionally, we prepared 50, 100, 150, and 200 nm thick α -MnTe films with a patterned W electrode on the surface to investigate the dependence of the film thickness on the transition temperature of the MnTe film. First, we formed patterned W electrodes on as-deposited β -MnTe films. Next, the β -MnTe films were annealed up to 550 °C and showed a transition to a low-resistance α -phase (see Supplementary Fig. 12A).” (p10, line 32nd-36th in the revised text).

4. The following sentence was added in Methods, Memory device characterization:

“The temperature dependence of the resistance of the MnTe devices and the MnTe films with patterned W electrodes was also investigated using the two-point probe method in the range of 20 °C –370 °C under an Ar/5% H₂ mixed gas atmosphere with a heating rate of about 10 °C/min.” (p11, line 10th-12th in the revised text).

5. We added the following reference in the supplementary text (supplementary reference section):

Thanigaimani, V. & Angadi M. A. Thickness dependence of temperature coefficient of resistance and neel temperature in MnTe films. *J. Mater. Sci. Lett.* **12**, 1052-1056 (1993).

Comment 5:

The authors should give a more thorough overview of the various polymorphs that have been observed in MnTe and discuss the potential formation of a zinc blende phase. This phase has been synthesized using MBE (Chem. Phys. Lett., 2004, 387, 110-115). Looking at the XRD pattern in Figure 1 this structure could also provide a reasonable fit. It is suggested that the range of this XRD pattern is extended to higher angles.

Authors: Thank you very much for your advice. Actually, sphalerite as designated by γ phase means zinc-blende structure. As the reviewer pointed out, XRD result of the as-deposited MnTe film in Fig. 1 has a good agreement with that of a γ phase with a zinc blende structure because the peak position of $(002)_\beta$ is almost the same with that of $(111)_\gamma$. Fig. R5 shows the XRD pattern of the as-deposited MnTe film without W cap-layer in the 2θ range from 20° to 55° . All Bragg reflection peaks are derived from those of a β crystalline phase. This is a strong proof that the as-deposited MnTe film has a wurtzite β phase, not zinc-blende γ phase.

Fig. R5. XRD pattern of the as-deposited MnTe film (1 μm thick) without a W cap layer.

In response to the reviewer's comment, we have added the additional results and revised the text as follows:

1. We added additional results of the XRD pattern of the as-deposited MnTe film (1 μm in thickness) without a W cap-layer in Supplementary Section 1 and Supplementary Fig. 1 in the revised manuscript.
2. The following sentence was added in Methods, Preparation of MnTe films:
"A 1 μm thick MnTe film without a W cap layer was also prepared to determine the crystal structure of the as-deposited film." (p9, line 25th-26th in the revised text).

3. We added the following sentence:

“We also confirmed via XRD measurements in the 2θ range from 20° to 55° that the as-deposited MnTe without a W cap layer has a β -phase (see Supplementary section 1 and Fig. 1).” (p3, line 16th-17th in the revised text).

Comment 6:

In addition, the authors should provide a more complete description of the state of the art in resistive memory switching research. It is certainly valuable that a comparison with a benchmark material Ge-Sb-Te is provided, but in terms of switching speed, there are certainly better/faster alternatives, such as HfO₂.

Authors: Thank you very much for your comment. We mentioned about a fast resistive switching memory of HfO₂. Actually, it has been reported that a HfO₂ resistive switching memory shows a fast operating speed of around (~ 5 ns)¹⁷.

In compliance with reviewer’s comment, we conducted the following modification:

1. We added the following sentence:

“This fast operation speed of the MnTe device is almost comparable to that of HfO₂ resistive switching memory (~ 5 ns)¹⁷.” (p4, line 13h-15th in the revised text).

2. We added the following reference:

17 Lee, H. Y., Chen, P. S., Wu, T. Y., Chen, Y. S., Wang, C. C., Tzeng, P. J., Lin, C. H., Chen, F., Lien, C. H. & Tsai, M. J. Low Power and High Speed Bipolar Switching with a Thin Reactive Ti Buffer Layer in Robust HfO₂ Based RRAM. IEEE International Electron Devices Meeting, 1-4 (2008).

Comment 7:

With respect to the durability of the devices, concerns can arise if one of the states is constituted of a metastable polymorph. In the present study durability is demonstrated over 400 cycles. Comparable studies using other resistive switching materials demonstrate much higher cycle numbers (often >100000). It would be of interest to investigate why the MnTe-based devices failed after 400 cycles and discuss how this issue could be solved.

Authors: Thank you very much for your comment. In compliance with the reviewer’s comment, we observed the cross-sectional TEM microstructure of the MnTe device which broke down after SET-RESET operation to understand the failure mechanism of the present device. Fig. R6 shows the cross-sectional bright-field STEM image and STEM-EDS images of the device. We found that there are cracks in the top W electrode at the contact hole, since the upper W electrode has a concave shape. We

also found that Mn diffuses out through the cracks and reacts with O to form MnO_x on the surface. The Mn diffusion out to the surface causes the composition deviation of the MnTe memory layer which leads the formation of MnTe₂ and voids. Such phase separation and void formation degrade the cyclic durability of the device. We confirmed that the Mn diffusion out to the surface dose not happen in other flat part without cracks in the top W electrode. This result indicates we need to modify the device structure to investigate the true cyclic durability of the MnTe memory layer. We believe that much better cyclic endurance can be obtained in the conventional mushroom-type device structure without any cracks in the electrode layer, as shown in Fig. R7.

Fig. R6. (A) Cross-sectional bright-field STEM image and the corresponding STEM-EDS images of Mn (purple), Te (light blue), W (red), Si (yellow), and O (green) of the MnTe device, which broke down after SET–RESET operation. We found that there are cracks in the top W electrode at the contact hole, since the upper W electrode has a concave shape. (B) Composite STEM-EDS image of Mn (purple) and W (red) indicating Mn diffused out to the surface through the crack in the W layer.

Fig. R7. A conventional mushroom-type device structure.

In response to the reviewer’s comment, we have added the additional results and revised the text as follows:

1. We added additional results of the TEM images of the MnTe device which broke down after SET–RESET operation in Supplementary Fig. 7.

2. The following sentence was added in Methods, Memory device characterization:

“To understand the failure mechanism of the MnTe device, we observed its cross-sectional TEM microstructure, which broke down after SET–RESET operation. The TEM sample was thinned using the FIB technique.” (p11, line 28th-30th in the revised text).

3. We added the following sentence:

“This cyclic durability is very limited compared to that observed in other memory devices (often >100,000)¹⁹. We found that the Mn in the MnTe layer diffuses onto the surface through cracks at the contact hole of the top W electrode and reacts with O to form MnO_x on the surface during cyclic operations. The diffusion of Mn out to the surface leads to the formation of MnTe₂ and voids in the MnTe layer, which degrades the cyclic durability (see Supplementary Fig. 7). The aforementioned cracks were unintentionally introduced during the deposition of the W layer because the contact hole has a concave shape. Further studies are needed to evaluate the cyclic durability of the MnTe film using a device structure without any cracks.” (p4, line 32nd - p5, line 7th in the revised text).

4. We added the following reference:

19 Athmanathan, A., Stanisavljevic, M., Papandreou, N., Pozidis, H. & Eleftheriou, E. Multilevel-Cell Phase-Change Memory: A Viable Technology. *IEEE Journal on Emerging and Selected Topics in Circuits and Systems* **6**, 87-100 (2016).

Comment 8:

It is also important, to assess the stability of the beta' phase without cycling. How long can the system remain in the beta' phase considering environmental factors (e.g. temperatures in the range of 70°C-80°C) without showing a transformation into the alpha ground state?

Authors: Thank you very much for your comment. In order to investigate the thermal stability of the metastable β' phase in the MnTe, we need to evaluate the activation energy of the phase transition from the β' to α phase. The activation energy can be estimated by investigating the dependence of the heating rate on the transition temperature from the β' to α phase (Kissinger plot). To do it, we need to prepare several MnTe devices with exactly the same structure to obtain a certain amount of the β' phase which is thermal stress-induced by RESET operation. However, the MnTe devices were fabricated using conventional photolithography combined with FIB drilling which causes some differences in the contact hole size, such as contact area, contact depth and so on, between fabricated devices. Even such small differences would affect the transformation temperature which is determined by the two-point probe method during heating.

Fig. R8. (A) Temperature dependence of the resistance of as-deposited MnTe films at various heating rates. (B) Kissinger plots for the activation energy of the β - to α -phase transition. From the slope of the plots, the activation energy was estimated to be 2.02 eV. (C) Plot of failure time vs. temperature for the MnTe film, which was obtained using the Ozawa method. Failure time was defined as the time when the transition started. (D) Resistance change as a function of annealing time at 200°C in a RESET-operated MnTe device (RESET operation: 4.2 V for 50 ns).

In this study, we used the activation energy of the transition from the metastable β to stable α phase to evaluate the thermal stability of the metastable β' phase because of the structural similarity between the β and the β' phases. Figure R8A shows the temperature dependence of the as-deposited MnTe films at various heating rates. The inset shows the transition temperature as a function of heating rate. The transition temperature increases with increasing heating rate. Figure R8B shows the Kissinger plots for the activation energy of the transition of the β to α phase in the MnTe film. The activation energy was estimated to be 2.02 eV. Based on the Ozawa method using the activation energy of 2.02 eV and the transition temperature of 227°C which was obtained in the MnTe device after RESET operation, the MnTe memory layer is expected to show a 10-year lifetime at a maximum at a maximum temperature of 103°C , as shown in Fig. R8C. However, this value was estimated using the activation energy for the transition from the β to α phase, not the transition from the β' to α phase. Therefore, we evaluated the thermal stability of a MnTe device with RESET state at 200°C . Figure R8D shows the resistance change as a function of annealing time at 200°C in a MnTe device after RESET operation. The device resistance decreases and increases during heating and cooling processes because of the

semiconductor nature of the MnTe film, but the device does not show a clear change in resistance with phase transition during the annealing at 200°C for 1h, indicating that the β' phase shows a good thermal stability at 200°C for over 1h. This result suggests that the thermal stability of the β' is much better than the expected one, as shown in Fig. R8C.

In response to the reviewer's comment, we have added the additional results and revised the text as follows:

1. We added additional results in Supplementary Section 10 and Supplementary Fig. 13.
2. The following sentence was added in Methods, MnTe film characterization:
“The dependence of the heating rate on the transition temperature of the as-deposited MnTe film was investigated using the two-point probe method at various heating rates of 4.6–23 °C/min, and the activation energy of the transition was estimated using the Kissinger method.” (p10, line 1st-3rd in the revised text).
3. We modified the original sentence (p4, line 28th-31th in the original text) The following sentence was added in the revised manuscript:
“As mentioned above, the transition temperature from the metastable β' -phase to the stable α -phase in the MnTe device was higher than that of the amorphous-to-crystalline phase change in conventional PCMs, 227 °C vs ~160 °C in GST225, indicating that the MnTe device has an adequate data retention ability (see Supplementary Fig. 11). We confirmed that the metastable β' -phase obtained via the RESET operation can be maintained at 200 °C for over 1 h. This thermal stability was much better than that estimated (for ~15 min at 200 °C) based on the Ozawa method²⁴ using the activation energy (2.02 eV) of the transition from the β -phase to the α -phase. From the estimation, the MnTe memory layer was estimated to show a 10-year lifetime at a maximum temperature of 103 °C. These results indicate that the MnTe device shows better data retention compared to the conventional GST225 device²⁵ (over 10 years at 85 °C) (see Supplementary section 10 and Fig. 13).” (p7, line 23rd-33rd in the revised text).

Comment 9:

Finally, in the conclusion multiple other developments in the field are listed. Of course this highlights the relevance and impact of the current study, but I would rather expect to find this information in the introduction. Instead I would appreciate if the authors could share their vision on how research on MnTe ReRAM devices should carry on and what needs to be done to make MnTe ReRAM devices a reality.

Authors: Thank you very much for your comment. In accordance with the reviewer's comment and also the formatting style of the journal, the following modifications were conducted:

1. We added the following sentence with reference which was suggested by the reviewer and words in the revised manuscript:

“Polymorphism is of great interest in many fields, including medicine, materials science, catalysis, and electronics, because different forms exhibit dramatically different physical and chemical properties³.” (p2, line 13th-15th in the revised text).

3. Gentili, D., Gazzano, M., Melucci, M., Jones, D. & Cavallini, M. Polymorphism as an additional functionality of materials for technological applications at surfaces and interfaces. *Chem. Soc. Rev.* **48**, 2502-2517(2019).

“This suggests that MnTe can exhibit a drastic change in electrical and optical properties through polymorphic change.” (p2, line 21st-22nd in the revised text).

“enables polymorphic change,” (p2, line 24th in the revised text).

2. We put the sentences referring to other application of MnTe in introductive part as follows:

“The present MnTe polymorphic-change film can be used in a single-polycrystalline structure fabricated by simple, conventional sputtering techniques. This material, therefore, should be a promising material as a memory layer in not only nonvolatile memory, but also photonic memory^{8,9}, nanodisplays^{10,11}, and neuromorphic devices^{12,13}.” (p3, line 5th-9th in the revised text).

3. We added the following sentence in compliance with the reviewer’s suggestion:

“To realize such a hypothesis, we need further studies on the optimal memory cell structure, optimal electrode material, and optimal selector material for the MnTe memory layer.” (p8, line 27th-28th in the revised text).

Comment 10:

There have been multiple studies on the magnetron sputtering of MnTe. In almost all of those reports (e.g. Reference 6, J. Mater. Chem. C, 2018,6, 6297-6304) the alpha phase (NiAs-Structure) was observed. More details regarding the geometry of the deposition chamber as well as base pressure, power densities should be given. The authors should also provide an estimate/measurement of the effective temperature during deposition since room temperature depositions are always subject to unintentional heating and the polymorph formation can be highly influenced by the substrate temperature during sputtering.

Authors: Thank you very much for your comment. We described the sputtering deposition in more detail in experimental method section.

In response to the reviewer’s comment, the following modifications were conducted:

1. We added the following sentences in the experimental method section; Preparation of MnTe films:
“, where the substrate holder was rotated during deposition. The diameter and thickness of the

targets were 50.8 mm and 5 mm, respectively. The distance between the target and the substrate was about 500 mm. The RF power used for sputtering deposition was 40–50 W for the Mn target and 10 W for the Te target.” (p9, line 7th-10th in the revised text).

“The working pressure was around 3.6×10^{-1} Pa. During the film deposition, the substrate was not intentionally heated, and no bias was applied to the substrate. The substrate temperature was monitored using a thermocouple attached to the substrate holder and was confirmed to be below 300 K (27 °C) during deposition.” (p9, line 12th-15th in the revised text).

Comment 11:

The starting point for the experiments as reported is a beta-MnTe, which is then heated to achieve the ground state alpha-MnTe. This alpha-phase is the thermodynamic ground state and would be the starting point for many other researchers using other sputter systems or near-equilibrium deposition techniques. The authors should discuss the impact of the structural properties of the starting materials. If the MnTe was sputtered at high-temperature (e.g. 500°C) the as deposited material would likely be already in the alpha phase. Does such a material also exhibit the discussed displacive phase transformation behaviour or is the starting point (a metastable state which is then annealed) important for the functionality of the device.

Authors: Thank you very much for your comment. In our additional experiments, we confirmed that the α phase obtained by annealed at around 300°C transforms to the β' phase just by heating which introduces a certain thermal stress in the MnTe film with patterned electrodes. This result indicates that the α phase obtained by sputtering deposition at high-temperature should also show such a displacive transformation under a certain thermal stress.

Comment 12:

On page 3 a calculation of the total operation energy is carried out. To relate these values to the transition temperatures discussed elsewhere in the manuscript the dimensions of the active region of device should be known. It is my understanding that this calculation gives the total energy whereas the local heating, which is relevant for the phase transformations very much depends on the geometry of the device.

Authors: Thank you very much for your comment. From the TEM images of the operated device, the active region size is about 500 nm in diameter and 100 nm in thickness, as shown in Fig. 2 in the original text. And, the phase transition temperature from the β' phase to the α phase which occurs in the active region was determined to be about 227°C (Supplementary Fig. 11).

As the reviewer pointed out, the phase transformation behavior, especially, transformation volume (~ active region) should be strongly influenced by the geometry of the device, causing a large variation of the total operation energy. Actually, it has been reported that the operation energy of memory devices

strongly depend on the size of the contact area between electrode and memory layer and it drastically decreases with decreasing the size of the contact area. Therefore, the operation energy of the MnTe device is also expected to be reduced by decreasing the contact area size between the MnTe and electrode.

We also conducted the additional experiment about the effect of the geometry of the device on the transformation temperature. As already shown in Fig. R2, R3 and R4, the obtained results indicate that the geometry of the device (device with contact hole fabricated by FIB technique (R2, R3), film with patterned electrode on the surface (R4), film thickness dependence (R4)) very slightly affects the transformation “starting” temperature which was determined by the resistance measurement during heating, as mentioned in the reply part for the comment 4.

In response to the reviewer’s comment, we added the following sentences:

1. “It has been reported that the operation energy strongly depends on the size of the contact area between the memory layer and the electrode, and this energy drastically decreases with the decrease of the size of that area¹⁸. This indicates that the operation energy of the MnTe device can be reduced by scaling the size of the contact area.” (p4, line 25th-28th in the revised text).

2. We added the following reference:

18 Lee, S., Sohn, J., Jiang, Z., Chen, H.-Y. & Wong H.-S. P. Metal oxide-resistive memory using graphene-edge electrodes. *Nature Commun.* **6**, 8407 (2015).

Minor corrections:

Typo on the bottom of page 3. “Frist”.

Authors: Thank you very much for your deep reading. We revised the word from “Frist” to “First”.

Reviewer #2:

Comment 1:

When referring to GST the authors probably refer to the GST225 composition and this should be clarified.

Authors: Thank you very much for your comment. The reviewer is correct. GST in the original manuscript refers to $\text{Ge}_2\text{Sb}_2\text{Te}_5$.

In accordance with the reviewer's comment, we replaced "GST" by "GST225", in the revised manuscript. And the sentences in the original manuscript (p2, line 22nd-26th in the original text) was revised as follows:

"A resistance (R) vs. voltage (V) curve under a 50 ns wide voltage pulse for a MnTe device is shown in Fig. 1B, in which a W layer was used for the top and bottom electrodes (see Supplementary Fig. 3). For comparison, the R - V curve for a typical phase-change memory device using $\text{Ge}_2\text{Sb}_2\text{Te}_5$ (GST225) PCM is also shown in Fig. 1B. Here, we observed a resistive switching behavior in the MnTe device, as with the GST225 device." (p4, line 4th-8th in the revised text)

Comment 2:

The cycling of the MnTe device is very low: only 400 cycles. For GST usually cycling is in the order of a million. The authors should explain the possible failure mechanisms and the possible future possibility to improve in that respect. Without the expectation of improving, MnTe is not at all suitable for any kind of memory. Thus, the manuscript does not deserve a publication in a high impact journal.

Authors: Thank you very much for your comment. As already mentioned above (in the reply section for the reviewer#1, comment 6), we observed the cross-sectional TEM microstructure of the MnTe device which broke down after SET-RESET operation to understand the failure mechanism of the present device. Figure R6 (shown in the reply section for the reviewer#1) shows the cross-sectional bright-field STEM image and STEM-EDS images of the device. We found that there are cracks in the top W electrode at the contact hole, since the upper W electrode has a concave shape. We also found that Mn diffuses out through the cracks and reacts with O to form MnO_x on the surface. The diffusion out of Mn causes the composition deviation of the MnTe memory layer which leads the formation of MnTe_2 . Such a phase separation degrades the cyclic durability of the device. We confirmed that the Mn diffusion out to the surface does not happen in other flat part without cracks in the top W electrode. This result indicates we need to modify the device structure to investigate the true cyclic durability of the MnTe memory layer. We believe that much better cyclic endurance can be obtained in the conventional mushroom-type device structure without any cracks in the electrode layer, as shown in Fig. R7.

In response to the reviewer's comment, we have added the additional results and revised the text as follows:

1. We added additional results of the TEM images of the MnTe device which broke down after SET-RESET operation in Supplementary Fig. 7.

2. The following sentence was added in Methods, Memory device characterization:

“To understand the failure mechanism of the MnTe device, we observed its cross-sectional TEM microstructure, which broke down after SET–RESET operation. The TEM sample was thinned using the FIB technique.” (p11, line 28th-30th in the revised text).

3. We added the following sentence:

“This cyclic durability is very limited compared to that observed in other memory devices (often >100,000)¹⁹. We found that the Mn in the MnTe layer diffuses onto the surface through cracks at the contact hole of the top W electrode and reacts with O to form MnO_x on the surface during cyclic operations. The diffusion of Mn out to the surface leads to the formation of MnTe₂ and voids in the MnTe layer, which degrades the cyclic durability (see Supplementary Fig. 7). The aforementioned cracks were unintentionally introduced during the deposition of the W layer because the contact hole has a concave shape. Further studies are needed to evaluate the cyclic durability of the MnTe film using a device structure without any cracks.” (p4, line 32th - p5, line 7th in the revised text).

4. We added the following reference:

19 Athmanathan, A., Stanisavljevic, M., Papandreou, N., Pozidis, H. & Eleftheriou, E. Multilevel-Cell Phase-Change Memory: A Viable Technology. *IEEE Journal on Emerging and Selected Topics in Circuits and Systems* **6**, 87-100 (2016).

Comment3:

The explanation of the displacive transformation is given on page 4, but for helping the readers it would be helpful to summarize the results a bit before. Furthermore, figure 3D is rather difficult to understand, as the shuffling is not very intuitive.

Authors: Thank you very much for your advice. We summarized the results a bit on the displacive transformation in the introductory part. And, to help readers understand, we added lines and arrows in 3D-structure images of Fig 3D, as with 2D-structure images of Fig. 3.

In response to the reviewer’s comment, we conducted the following modifications:

1. We added the following sentence in the introductory part:

“The low-resistance α -phase (NC-type structure) transforms into a high-resistance phase with a WZ-type structure (β' -phase), which exhibits slight differences in the coordinates of the Te atoms in the β -phase. Shuffling occurs every two planes along the [210] and $[\bar{2}\bar{1}0]$ directions in the α -phase or the β' -phase, which corresponds to the minimum displacements to achieve the polymorphic change between the sixfold-coordinated NC-type and fourfold-coordinated WZ-type structures.” (p2, line 30th - p3, line 3rd in the revised text).

2. We revised Fig. 3D as follows:

Fig. R9. Polymorphic-change mechanism between the α and β' phases through alternate atomic-plane shuffling. Upper images show a crystal structure of the α (left) and β' (right) phases. Middle images show a projection of atoms in the α (left) and β' (right) phases viewed from the $[001]$ direction. Bottom images present a projection of atoms on the light-blue colored plane in the α ($(\bar{1}20)$, left) and β' ($(\bar{1}20)$, right) phase structure. The dotted arrows indicate the shuffling direction of each atomic-plane for the polymorphic-change from the α to β' phase (left) and from the β' to α phase (right).

Comment4:

A recent paper on PCM superlattices (Boniardi et al. Phys. Status Solidi RRL 13 (4), 1800634 (2019) <https://doi.org/10.1002/pssr.201970021>) has been also reporting low power consumption it might be useful to cite such work to compare with it and not with standard GST.

Authors: Thank you very much for your comment. We added the reference in the revised manuscript.

1. We added the following reference:

27 Boniardi, M., Boschker, J. E., Momand, J., Kooi, B. J., Redaelli, A. & Calarco, R. Evidence for Thermal-Based Transition in Super-Lattice Phase Change Memory. *Phys. Status Solidi RRL* 13 (4), 1800634 (2019).

Reviewer #3:

Comment 1:

In GSTs the contrast of the properties (electrical, optical) caused by the difference in the chemical bonding in amorphous and the metastable rocksalt crystalline state. GSTs have stable hexagonal phase whose properties are different from rocksalt crystalline state. Actually, the amplitude of the properties contrast is driven by the interplay of effects of the resonant bonding and the disorder. In the manuscript Authors demonstrated the difference between different crystalline states of MnTe and therefore claim that this material might be promising for new memory concepts. It is, however, need to be clarified in details what is the origin of the property contrast in different crystalline states of MnTe and to discuss the possible “tools” (like disorder in GSTs) that might manage the amplitude of the properties contrast.

Authors: Thank you very much for your comment. The contrast of electrical and optical properties upon the phase transition is caused by the change of the coordination number of Mn (or Te) atoms, which affects the hybridization between Mn *d*-states and Te *p*-states. As mentioned in the original manuscript, the α phase has a NiAs-type hexagonal (NC-type) structure and the coordination number of Mn (or Te) atoms is 6, namely, a six-fold coordinated NC-type structure, while the β (or β') phase has a wurtzite-type hexagonal (WZ-type) structure and the coordination number of Mn (or Te) atoms is 4, namely, a four-fold coordinated WZ-type structure. Siol et al. have recently calculated the density of states of Mn-chalcogenides, such as MnTe, MnSe and MnSe_{0.5}Te_{0.5}, with different structures. They demonstrated that the band gap of a four-fold coordinated WZ-type structure (2.71 eV) is wider than that of a six-fold coordinated NC-type structure (0.98 eV) because of the weaker hybridization between Mn *d*-states and Te *p*-states in four-fold coordination (p2, line 11th-16th in the original text).

To make this clear, we modified the original sentences as follows:

1. The old sentences (p2, line 11th-20th in the original text) was revised as follows:

“Based on the Hall-effect measurements at room temperature (RT), we also found that the drastic resistivity drop upon polymorphic change is mainly caused by a drastic increase in the carrier density (see Supplementary Table 1). Our optical measurements indicated that the NC-type α -phase (1.48 eV) has a narrower bandgap compared to the WZ-type β -phase (2.50 eV) (see Supplementary Table 1). Siol et al. have recently calculated the density of states of Mn chalcogenides, such as MnTe, MnSe, and MnSe_{0.5}Te_{0.5}, with different structures. They demonstrated that the bandgap of a fourfold-coordinated WZ-type structure (2.71 eV) is wider than that of a sixfold-coordinated NC-type structure (0.98 eV) because of the weaker hybridization between the Mn *d*-states and Te *p*-states in the fourfold coordination¹⁶. That is, the drastic decrease in resistance upon the phase transition from the β -phase to the α -phase is considered to be due to a decrease in the coordination number of Mn (or Te) atoms from 6 to 4, which weakens the hybridization between the Mn *d*-states and Te *p*-states.” (p3, line 21st-33rd in the revised text).

2. The old sentences (p4, line 14th-16th in the original text) was revised as follows:

“This transition pathway corresponds to the minimum displacements, in order to achieve the polymorphic change between the sixfold-coordinated NC-type and the fourfold-coordinated WZ-

type structures, which causes a large resistance contrast.” (p6, line 18th-20th in the revised text).

Comment 2:

To be employed in the memory device on one hand material need to show the stability of the states attributed to the logical “0” and “1”, on the other hand need to show reasonable quantity of writing cycles. In case of GSTs, these parameters are in the scales of years and millions of cycles, correspondingly. In the manuscript Authors write about “over 400 switching cycles”, that is definitely not enough for real industrial applications. What is the maximum possible number switching cycles in case of MnTe? What about the stability of corresponding crystalline states attributed to the logic “0” and “1”?

Authors: Thank you very much for your comment. The reviewer pointed out, at the moment, the MnTe device showed a poor cyclic durability for actual practical application. As mentioned above (our reply parts for the reviewer#1, comment 6 and reviewer#2, comment 2), we found that the poor cyclic durability of our MnTe devices is due to crack defects in the top W electrode at the contact hole, as shown in Fig. R6. We also found that Mn diffuses out through the cracks and reacts with O to form MnO_x on the surface. The diffusion out of Mn causes the composition deviation of the MnTe memory layer which leads the formation of MnTe₂. Such a phase separation degrades the cyclic durability of the device. We confirmed that the Mn diffusion out to the surface dose not happen in other flat part without cracks in the top W electrode. This result indicates we need to modify the device structure to investigate the true cyclic durability of the MnTe memory layer. We believe that much better cyclic endurance can be obtained in the conventional mushroom-type device structure without any cracks in the electrode layer, as shown in Fig. R7.

In response to the reviewer’s comment, we have added the additional results and revised the text as follows:

1. We added additional results of the TEM images of the MnTe device which broke down after SET-RESET operation in Supplementary Fig. 7.
2. The following sentence was added in Methods, Memory device characterization:
“To understand the failure mechanism of the MnTe device, we observed its cross-sectional TEM microstructure, which broke down after SET–RESET operation. The TEM sample was thinned using the FIB technique.” (p11, line 28th-30th in the revised text).
3. We added the following sentence:
“This cyclic durability is very limited compared to that observed in other memory devices (often >100,000)¹⁹. We found that the Mn in the MnTe layer diffuses onto the surface through cracks at the contact hole of the top W electrode and reacts with O to form MnO_x on the surface during cyclic operations. The diffusion of Mn out to the surface leads to the formation of MnTe₂ and voids in the MnTe layer, which degrades the cyclic durability (see Supplementary Fig. 7). The aforementioned

cracks were unintentionally introduced during the deposition of the W layer because the contact hole has a concave shape. Further studies are needed to evaluate the cyclic durability of the MnTe film using a device structure without any cracks.” (p4, line 32th - p5, line 7th in the revised text).

4. We added the following reference:

19 Athmanathan, A., Stanisavljevic, M., Papandreou, N., Pozidis, H. & Eleftheriou, E. Multilevel-Cell Phase-Change Memory: A Viable Technology. *IEEE Journal on Emerging and Selected Topics in Circuits and Systems* **6**, 87-100 (2016).

Moreover, we discussed about the stability of the α phase and the β' phase, as mentioned in the reply part for the reviewer#1. We found by the resistance measurement of the MnTe device upon heating that the high resistance β' phase (“0”) transforms to the low resistance α phase (“1”) at around 220°C upon heating, while the low resistance α phase transforms to the high resistance β' phase at around 300°C upon heating, as shown in Fig. R3 (the same with Fig. 4 in the revised manuscript). These transition temperatures are sufficiently higher than the crystallization temperature of conventional GST225 (~160°C), indicating that the MnTe device shows better thermal stability. Generally, thermal stability of memory materials is discussed by estimating a maximum temperature ensuring a 10-year lifetime. Therefore, we tried to investigate the maximum temperature for 10-year lifetime of the metastable β' phase in the MnTe, as mentioned in the reply part for the reviewer#1, comment 7. The MnTe memory layer is expected to show a 10-year lifetime at a maximum temperature of 103°C. These results indicate that the MnTe device shows better endurance than conventional GST225 device (~85°C).

In response to the reviewer’s comment, we have added the additional results and revised the text as follows:

1. We added additional results in Supplementary Section 10 and Supplementary Fig. 13.

2. The following sentence was added in Methods, MnTe film characterization:

“The dependence of the heating rate on the transition temperature of the as-deposited MnTe film was investigated using the two-point probe method at various heating rates of 4.6–23 °C/min, and the activation energy of the transition was estimated using the Kissinger method.” (p10, line 1st-3rd in the revised text).

3. We modified the original sentence (p4, line 28th-31th in the original text) The following sentence was added in the revised manuscript:

“As mentioned above, the transition temperature from the metastable β' -phase to the stable α -phase in the MnTe device was higher than that of the amorphous-to-crystalline phase change in conventional PCMs, 227 °C vs ~160 °C in GST225, indicating that the MnTe device has an adequate data retention ability (see Supplementary Fig. 11). We confirmed that the metastable β' -phase obtained via the RESET operation can be maintained at 200 °C for over 1 h. This thermal stability was much better than that estimated (for ~15 min at 200 °C) based on the Ozawa method²⁴ using

the activation energy (2.02 eV) of the transition from the β -phase to the α -phase. From the estimation, the MnTe memory layer was estimated to show a 10-year lifetime at a maximum temperature of 103 °C. These results indicate that the MnTe device shows better data retention compared to the conventional GST225 device²⁵ (over 10 years at 85 °C) (see Supplementary section 10 and Fig. 13).” (p7, line 23rd-33rd in the revised text).

4. We added the following references:

24 Ozawa, T. Kinetic analysis of derivative curves in thermal analysis. *J. Therm. Anal.* **2**, 301-324 (1970).

25 Raoux, S., Welnic, W. & Ielmini, D. Phase Change Materials and Their Application to Nonvolatile Memories. *Chem. Rev.* **110**, 240-267 (2010).

Comment 3:

If in the Abstract Authors write “...we show a MnTe semiconductor film that exhibits a reversible displacive transformation that results in large electrical and optical contrasts..”, I will strongly recommend to improve the part discussing the optical contrast by providing corresponding data in wide spectral range that will definitely support the mentioned above statement.

Authors: Thank you very much for your comment. As the reviewer pointed out, the optical contrast in wide spectral range is important for e.g., photonic device application. At the moment, we cannot measure the optical characteristics of the β' phase because we cannot obtain a large size area of the β' phase for the optical measurement, such as its reflectance and transmittance as a function of wavelength. Therefore, we measured the optical spectra of the α phase and the β phase which has very similar four-fold coordinated structure in the wavelength range between 400 nm and 1100 nm. We added the additional results on optical contrast in wide spectral range, as shown in Fig R9. Figure R9A shows the reflectance curves as a function of wavelength obtained in 150-nm thick MnTe films on the SiO₂/Si substrate, while Figure R9B shows transmittance curves as a function of wavelength obtained in a 100-nm thick MnTe films on the glass substrate. We found that the reflectance of the β -MnTe film on the SiO₂/Si substrate is lower than that of the α -MnTe film on the SiO₂/Si substrate in the wavelength range between 800 and 900 nm, while in other wavelength range, the α -MnTe film shows lower reflectance than the β -MnTe film. The transmittance was higher in the β -MnTe film on the glass substrate than in the α -MnTe film on the glass substrate in the whole wavelength range.

Fig. R10 (A) Reflectance curves as a function of wavelength obtained in 150 nm thick MnTe films on a SiO₂/Si substrate. (B) Transmittance curves as a function of wavelength obtained in 100 nm thick MnTe films on a glass substrate.

In response to the reviewer's comment, we conducted the following modifications:

1. We added the additional results in Supplementary Section 11 and Supplementary Fig. 14.
2. We added the following sentences:
 “We confirmed that the reflectance of the β -MnTe film on the SiO₂/Si substrate is lower than that of the α -MnTe film on the SiO₂/Si substrate in the wavelength range between 800 and 900 nm, whereas in the other wavelength range, the α -MnTe film shows lower reflectance compared to the β -MnTe film. The transmittance was higher in the β -MnTe film on a glass substrate than in the α -MnTe film on a glass substrate for the entire wavelength range (see Supplementary Section 11 and Supplementary Fig. 14).” (p8, line 7th-13th in the revised text).

Other corrections:

1. We revised the formatting style, in accordance with the journal style, Introduction, Results, Discussion etc.
2. The abstract and the sentences in the old manuscript (p1, the beginning of the text – p2, line 3rd) was modified as described in the introductory part of the revised manuscript. (p1, line 8th-21st)
3. We added the following notes for supplementary information:
 (see Supplementary section 3) (p4, line 3rd-4th)
 (see Supplementary section 4) (p4, line 21st-22nd)
4. We also added the following sentences and references in the discussion section to cite recent related studies:
 “In addition, the MnTe film showing thermal-stress induced polymorphic-change should be very attractive material for straintronics in which a phase transition induced by mechanical strain

causes a change in electrical or optical properties³³⁻³⁴.” (p8, line 29th-31th in the revised text).

33 Ong, M. T. & Reed, E. J. Engineered Piezoelectricity in Graphene. *ACS nano*. **6**, 1387-1394 (2012).

34 Hou, W., Azizimanesh, A., Sewaket, A., Peña, T., Watson, C., Liu, M., Askari, H. & Wu, S. M. Strain-based room-temperature non-volatile MoTe₂ ferroelectric phase change transistor. *Nat. Nanotech.* **14**, 668-673 (2019).

5. We revised from “mm” to “nm” in Methods, preparation of MnTe films. (p9, line 18th)
6. We revised the captions in Fig. 3C as “(C) SADPs from the matrix enclosed by the red-dotted line (upper) and active region enclosed by the blue-dotted line (bottom).” In the original manuscript, “matrix” and “active region” were opposite in this sentence.
7. We revised from “Nature” to “*Nature*” in the Reference.
8. We added the following note about another Grant Number in the Acknowledgments:
and 19J21117
9. We revised from “Y.Ssutou” to “Y.Sutou” in the Author contributions.

Reviewers' comments:

Reviewer #1 (Remarks to the Author):

The authors performed a very detailed and thorough revision of the manuscript. All concerns and comments have been fully addressed and additional experimental results are provided where necessary. The additional results not only further support the claims and conclusions made in the original manuscript but also increase the reproducibility of the results by other researchers working on the same or related material systems.

I think that this revised version of the manuscript is now suitable for publication in Nature Communications.

Reviewer #2 (Remarks to the Author):

The manuscript is really well written and based on a thorough analysis of the data. Furthermore, all the issues raised by the different referees are addressed seriously and with a large amount of new measurements and supporting data, thus now all the conclusions are well supported by the experimental results. Based on these considerations and the importance of the present work I am of the opinion that the present manuscript is well suited for publication in Nature Communications and it can be published in the present form.

Reviewer #3 (Remarks to the Author):

All my comments were taken into account by Authors.
I would like to recommend the revised version of the manuscript for the publication in Nature Communications.

[Response to the referees' comments]

Reviewer #1 (Remarks to the Author):

The authors performed a very detailed and thorough revision of the manuscript. All concerns and comments have been fully addressed and additional experimental results are provided where necessary. The additional results not only further support the claims and conclusions made in the original manuscript but also increase the reproducibility of the results by other researchers working on the same or related material systems. I think that this revised version of the manuscript is now suitable for publication in Nature Communications.

Authors: Thank you very much for your comments. We're so pleasure to hear that the manuscripts were revised satisfactory for all points as you pointed out. We appreciate your deep reading and review.

Reviewer #2 (Remarks to the Author):

The manuscript is really well written and based on a thorough analysis of the data. Furthermore, all the issues raised by the different referees are addressed seriously and with a large amount of new measurements and supporting data, thus now all the conclusions are well supported by the experimental results. Based on these considerations and the importance of the present work I am of the opinion that the present manuscript is well suited for publication in Nature Communications and it can be published in the present form.

Authors: Thank you very much for your comments. We're grateful for your mention about the issues raised by the different reviewers as well as the comments you have pointed out. We appreciate your kind review.

Reviewer #3 (Remarks to the Author):

All my comments were taken into account by Authors. I would like to recommend the revised version of the manuscript for the publication in Nature Communications.

Authors: Thank you very much for your comment. We're so glad to hear that you have agreement with the revised manuscripts. And also, thank you very much for your recommendation for the publication in Nature Communications.

[Response to editorial requests]

1. We encourage increased transparency in peer review by publishing the reviewer comments and author rebuttal letters of our research articles, if the authors agree. Such peer review material is made available as a supplementary peer review file. Please state in the cover letter 'I wish to participate in transparent peer review' if you want to opt in, or 'I do not wish to participate in transparent peer review' if you don't. Failure to state your preference will result in delays in accepting your paper for publication.

Please note: we allow redactions to authors' rebuttal and reviewer comments in the interest of confidentiality. If you are concerned about the release of confidential data, please let us know specifically what information you would like to have removed. Please note that we cannot incorporate redactions for any other reasons. Reviewer names will be published in the peer review files if the reviewer signed the comments to authors, or if reviewers explicitly agree to release their name. For more information, please refer to our FAQ page at: <https://www.nature.com/documents/ncomms-transparent-peer-review.pdf>

Authors: We wish to participate in transparent peer review.

2. Please ensure that an updated editorial policy checklist that verifies compliance with all required editorial policies is completed and uploaded with the revised article. All points on the policy checklist must be addressed; if needed, please revise your manuscript in response to these points. Please note that this form is a dynamic 'smart pdf' and must therefore be downloaded and completed in Adobe Reader, instead of opening it in a web browser. Editorial policy checklist: <https://www.nature.com/documents/nr-editorial-policy-checklist.pdf>

Authors: We have confirmed all points provided in this checklist and also submitted the answered file.

3. Your manuscript should comply with our policies and format requirements, detailed in our checklist for authors at: <https://www.nature.com/documents/ncomms-manuscript-checklist.pdf>

Authors: We have checked all points provided in this checklist and filled in all blanks. In compliance with the policies in this document, we have performed some revisions indicated in "Our modification" section below.

4. Please also review the changes in the attached copy of your manuscript, which has been edited for style, and address the comments and queries I have added. If using Word, please use the 'track changes' feature to make the process of accepting your manuscript more efficient.

Authors: Thank you very much for your kindest mention for revision. We have performed some revisions in accordance with your requests, which are also indicated in "Our modification" section below. We also add our brief reply to each comment in attached PDF file one by one.

5. Data availability statements and data citations policy: All Nature Communications manuscripts must include a section titled "Data Availability" as a separate section after the Methods section but before the References. For more information on this policy, and a list of examples, please see <https://www.nature.com/documents/nr-data-availability-statements-data-citations.pdf>

Authors: We have confirmed it. Our manuscript contains "Data availability" section before Reference.

6. Your paper will be accompanied by a two-sentence editor's summary, of between 250-300 characters, when it is published on our homepage. Could you please approve the draft summary below or provide us with a suitably edited version.

Authors: Thank you very much for editing the summary. We approve the proposed summary.

[Our modifications]

The modified parts were highlighted by blue color in the main manuscript. For Supplementary information, we don't use the blue characters and number of lines because the PDF file is final version for publication.

At first, based on the policies provided in the checklist, we have performed the following corrections:

1. All unit dimensions described in both main manuscript and supplementary information were corrected in accordance with the checklist, e.g., $\text{m}\cdot\text{s}^{-1}$, $\text{ppm}\cdot\text{C}^{-1}$ or $\text{C}\cdot\text{min}^{-1}$.
2. The scale bar of all images in both main manuscript and supplementary information were fixed not to be labelled in figures. The length of scale bar was written in the legend of each figure.

Secondary, based on the editor's comment provided in attached PDF files, we have performed following corrections.

1. We revised the formatting style in the supplementary information, in accordance with the journal policy, Cover page, Supplementary Figures, Tables, Notes and References.
2. Supplementary References were fixed to be numbered sequentially from 1 to 4 in itself.
3. We have modified the abstract in the main manuscript. We added the introductory sentence in the first half of the abstract and modified the abstract so that it is less than 150 words. (p1, line 12th-13th)
4. When referring the supplementary information in the main manuscript, the words "Supplementary section" in the original text were replaced by the words "Supplementary Note" in the revised text. When we refer supplementary Figures, Tables and Note at the same time, they are also lined up in this order to follow the formatting style in supplementary information. In accordance with the modifications, we also fixed the sentence of "Supplementary information" section in the main manuscript.
5. The equation of the orientation relationship was numbered. (p6, line 11th)
6. The titles in each figure were revised to be shorter in both the main manuscript and supplementary information.
7. We added the definitions for abbreviations written in the captions of all figures for both main manuscript and supplementary information.

[Minor corrections]

We also added the following modifications in the main manuscript:

1. The divided items in each figure were fixed to be labeled with "a, b, c..." in both main manuscript and Supplementary information, like other papers published in Nature Communications.
2. We revised the words "Fig. S3" to the word "Supplementary Fig. 3". (p11, line 2nd)
3. We revised the words "Fig. 2A" to the word "Fig. 2b" in the caption of Fig. 3.
4. We added the following sentences in the Acknowledgement section: "We wish to thank Kosei Kobayashi, Takamichi Miyazaki, and Masatoshi Tanno, Tohoku University, Japan for the help with the TEM measurements; and Junichi Koike, Tohoku University, Japan for fruitful discussion." (p15, line 8th-10th)

5. Coordinate axes in atomic images of MnTe device (Fig. 3a, 5c and Supplementary Fig. 10) have been indexed as plane index. To avoid confusion, we corrected them to direction index to correspond with the schematic image in Fig. 3d.